# Sensorimotor pathway controlling stopping behavior during chemotaxis in the *Drosophila melanogaster* larva

**Ibrahim Tastekin[1,2†], Avinash Khandelwal[1,3], David Tadres[1,2,4,5], Nico D Fessner[1,2‡], James W Truman[3§], Marta Zlatic[3,6], Albert Cardona[3,7], Matthieu Louis[1,2,5,8]***

[1]EMBL-CRG Systems Biology Research Unit, Centre for Genomic Regulation, The Barcelona Institute of Science and Technology, Barcelona, Spain; [2]Universitat Pompeu Fabra, Barcelona, Spain; [3]Janelia Research Campus, Howard Hughes Medical Institute, Ashburn, United States; [4]Institute of Molecular Life Sciences, University of Zurich, Zurich, Switzerland; [5]Department of Molecular, Cellular and Developmental Biology & Neuroscience Research Institute, University of California, Santa Barbara, United States; [6]Department of Zoology, University of Cambridge, Cambridge, United Kingdom; [7]Department of Physiology, Development and Neuroscience, University of Cambridge, Cambridge, United Kingdom; [8]Department of Physics, University of California Santa Barbara, California, United States

**\*For correspondence:**
mlouis@lifesci.ucsb.edu

**Present address:** [†]Champalimaud Research, Champalimaud Center for the Unknown, Lisbon, Portugal; [‡]Institute for Molecular Biotechnology, NAWI Graz University of Technology, Graz, Austria; [§]Friday Harbor Laboratories, University of Washington, Washington, United states

**Competing interests:** The authors declare that no competing interests exist.

**Abstract** Sensory navigation results from coordinated transitions between distinct behavioral programs. During chemotaxis in the *Drosophila melanogaster* larva, the detection of positive odor gradients extends runs while negative gradients promote stops and turns. This algorithm represents a foundation for the control of sensory navigation across phyla. In the present work, we identified an olfactory descending neuron, PDM-DN, which plays a pivotal role in the organization of stops and turns in response to the detection of graded changes in odor concentrations. Artificial activation of this descending neuron induces deterministic stops followed by the initiation of turning maneuvers through head casts. Using electron microscopy, we reconstructed the main pathway that connects the PDM-DN neuron to the peripheral olfactory system and to the pre-motor circuit responsible for the actuation of forward peristalsis. Our results set the stage for a detailed mechanistic analysis of the sensorimotor conversion of graded olfactory inputs into action selection to perform goal-oriented navigation.
DOI: https://doi.org/10.7554/eLife.38740.001

## Introduction

Animals explore their environment to locate food while avoiding danger. This process is directed by the detection and the integration of multimodal cues, which organize the sequential release of behavioral programs. Evolution has produced a variety of orientation mechanisms responding to the anatomical and physiological constraints of each animal (*Fraenkel and Gunn, 1961*). These mechanisms share a common algorithmic basis: attractive cues promote motion toward favorable directions while aversive cues suppress it. This algorithm is exemplified in bacterial chemotaxis where bouts of straight motion ('runs') alternate with turns ('tumbles') that randomize the direction of motion (*Berg, 2008*; *Bi and Sourjik, 2018*). In spite of the undirected nature of reorientation through tumbling, bacteria efficiently ascend gradients of attractive chemicals (e.g., sugar) by extending their runs toward positive gradients. Similarly, the nematode *C. elegans* and the *D. melanogaster* larva suppress turning during up-gradient runs, but turning is facilitated during down-gradient runs (*Pierce-Shimomura et al., 1999*; *Lockery, 2011*; *Gomez-Marin and Louis, 2012*). Adult

flies walk upwind when stimulated by an attractive odor. Termination of odor stimulations elicits stops and reversals in walking adult flies (*DasGupta et al., 2014*; *Bell and Wilson, 2016*; *Álvarez-Salvado et al., 2018*). Flying insects such as pheromone-seeking male moths and foraging adult flies respond to attractive olfactory cues by implementing upwind 'surges'. Exiting the odor plume triggers searches through casting behavior (*Murlis et al., 1992*; *Bhandawat et al., 2010*; *van Breugel and Dickinson, 2014*). These examples illustrate the prowess of animals to modulate their turn rates based on the detection of changes in odor concentration. Whereas the biochemical pathway underlying the sensorimotor control of chemotaxis has been extensively studied in bacteria (*Bi and Sourjik, 2018*), little is known about the neural logic underlying the sensorimotor control of navigational behavior in animals with a nervous system.

Drosophila larvae display robust orientation behavior in response to food odors (*Cobb, 1999*). Larval chemotaxis results from coordinated transitions between four behavioral primitives: runs (forward motion by means of symmetrical peristaltic contractions), stops (cessation of peristaltic waves), head casts (lateral head sweeps) and turns (asymmetrical contractions of the body segments followed by straightening of the body) (*Green et al., 1983*; *Gomez-Marin et al., 2011*). To accurately navigate attractive odor gradients, larvae implement two sensorimotor tasks: they time the initiation of reorientation maneuvers based on the detection of negative sensory gradients (*Figure 1Ai*) and they direct turns toward the local odor gradient (*Gomez-Marin and Louis, 2012*). The sensorimotor algorithm underlying each of these tasks has been studied in controlled olfactory environments (*Louis et al., 2008*; *Gershow et al., 2012*) by exploiting computer-vision tools to correlate sensory inputs with elementary orientation responses (*Gomez-Marin et al., 2011*; *Gershow et al., 2012*). Acute activations of olfactory sensory neurons (OSNs) through optogenetics have established quantitative relationships between the activity of the peripheral olfactory system and the release of specific motor programs such as stops prior to the initiation of reorientation maneuvers (*Gepner et al., 2015*; *Hernandez-Nunez et al., 2015*; *Schulze et al., 2015*). Nonlinear aspects of the input-output response properties of the larval OSNs are essential to explain how dynamic patterns of stimulus intensity evoke predictable and stereotypical sequences of behavioral control (*Schulze et al., 2015*). In contrast with recent progress made in our understanding of the sensorimotor mechanisms directing larval chemotaxis, the neural circuits that convert the OSN activity into action selection — running, stopping, casting and turning — remain elusive. In the present work, we set out to reconstruct the sensorimotor pathway that controls the release of stopping behavior in response to the detection of negative changes in the concentration of an attractive food odor. Given the robust nature of larval chemotaxis, this work presents an opportunity to study the neural mechanisms that convert temporal changes in stimulus intensity into stereotyped behavioral sequences underlying goal-oriented behavior.

The larval olfactory system is compact: it comprises only 21 olfactory sensory neurons (OSN) (*Fishilevich et al., 2005*; *Kreher et al., 2005*). The axon terminals of OSNs synapse onto second-order projection neurons (PNs) in the antennal lobe (AL). The anatomical organization of the larval AL has been characterized through a combination of light and electron microscopy (*Ramaekers et al., 2005*; *Masuda-Nakagawa et al., 2009*; *Berck et al., 2016*). The axons of the PNs project to two main neuropil centers in the larval brain lobes where further sensory processing takes place: the lateral horn (LH) and the mushroom body (MB). The entire connectome of the larval MB has been reconstructed using electron microscopy (*Eichler et al., 2017*). In the adult *Drosophila*, PN axons arising from the same glomerulus of the AL converge onto stereotyped and overlapping regions of the LH (*Marin et al., 2002*; *Wong et al., 2002*). Distinct odors trigger calcium responses in specific regions of the LH in a way that is concentration-dependent (*Strutz et al., 2014*). Aside from these observations in adult flies, little is known about the function of the *Drosophila* lateral horn in the control of innate olfactory behaviors (*Dolan et al., 2018*).

At the other end of the central nervous system of the larva, pre-motor circuits located in the ventral nerve cord (VNC) are responsible for generating patterns of rhythmic motor activities that create peristaltic motion through stereotyped sequences of muscle contractions (*Inada et al., 2011*; *Berni et al., 2012*; *Kohsaka et al., 2012*; *Berni, 2015*; *Pulver et al., 2015*; *Fushiki et al., 2016*). Central pattern generating (CPG) networks underlie the production of rhythmic motor output in the absence of sensory feedback or input from other brain regions (*Marder and Calabrese, 1996*; *Katz, 2016*). The observation that isolated larval central nervous systems (CNS) are sufficient to produce segmentally-coordinated motor output suggests that the CPGs underlying peristalsis are

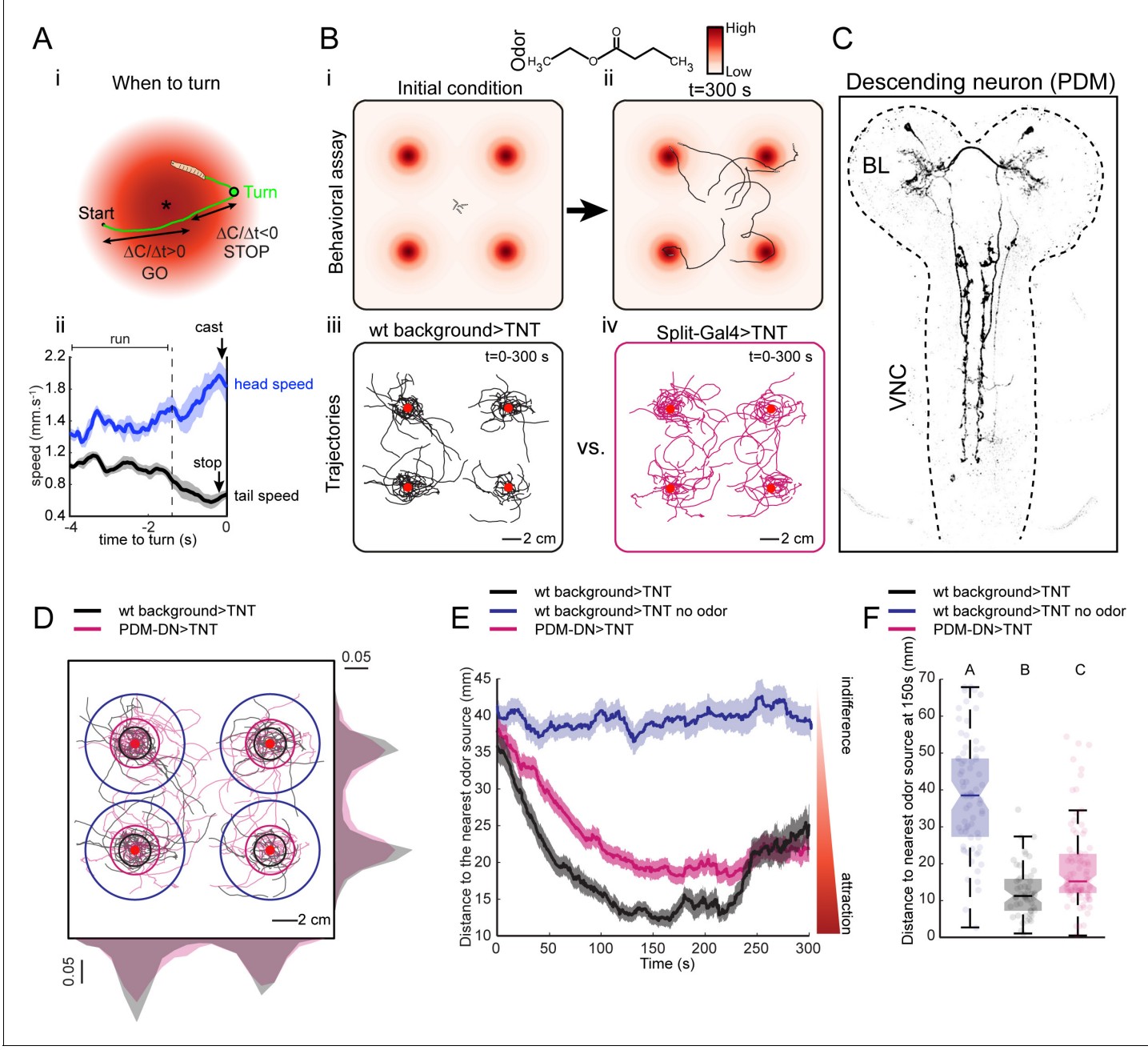

**Figure 1.** A loss-of-function behavioral screen using Split-Gal4 driver lines reveals a descending neuron that directs larval chemotaxis. (A) Sensorimotor model for *when-to-turn* decision during larval chemotaxis. (Ai) During up-gradient runs ($\Delta C/\Delta T > 0$), stopping and turning maneuvers are suppressed. By contrast, stops and turns are promoted during down-gradient runs ($\Delta C/\Delta T < 0$). (Aii) Turn-triggered averages of the head and tail speeds (10 turning events). Prior to reorientation, larvae stop (indicated by a drop in average tail speed) and sample their environment through lateral head casts (reflected by an increase in average head speed). Shaded areas indicate SEM. (B) Behavioral assay used in the loss-of-function screen. (Bi-ii) Assay description: about 20 larvae at the third instar developmental stage are placed in the middle of a large square Petri dish covered with 4% agar. Four-odor droplets of 8 μl (concentration: 15 mM) are placed at equidistant positions inside the lid of the Petri dish (*Figure 1—figure supplement 1*). The assay is overlaid with a color map representing the numerically simulated odor gradient (see Materials and methods). Larvae are tracked for 300 s. (Biii-iv) Trajectories for the wild-type control (left, black traces) and a group of Split-Gal4 >TNT larvae (right, magenta traces). Red dots indicate the position of the odor droplets. (C) Anatomy of the olfactory descending neuron identified in the loss-of-function screen. This descending neuron is denoted PDM-DN in reference to the location of its cell body (see *Figure 2*). (D) Representative trajectories of PDM-DN-silenced larvae (magenta) superimposed onto the trajectories of wild-type control (black). Distributions of centroid positions along the *x* and *y* axes are displayed next to the corresponding axis. PDM-DN-silenced larvae disperse more widely around the odor source compared to the wild-type control. Circles with different colors indicate the average distances of larvae to the nearest odor source (black: wild-type background with odor; blue: wild-type background without odor; magenta: PDM-

*Figure 1 continued on next page*

*Figure 1 continued*

DN>TNT). Red dots indicate the positions of the four odor droplets. (E) Time course of the distances between the centroids of groups of larvae and their nearest odor source. On average, larvae with a loss-of-function of PDM-DN (magenta) are less efficient at locating the odor sources as reflected by the larger distance to the odor source. Shaded areas indicate SEM. (F) Distance to the nearest odor source quantified at 150 s (duration of time window: 1 s). Each point in the background of the boxplot represents an independent trajectory. Horizontal lines represent the median of each sample. Semi-transparent boxes represent the 25th and 75th percentiles. The whiskers extend to the most extreme data points excluding the outliers (default settings of the 'boxplot' function of Matlab_R2015b). Different letters indicate statistically significant difference (Wilcoxon rank-sum test, p<0.05 upon Bonferroni correction). Number of trajectories tested: n = 71 for wild-type control with no odor, n = 53 for wild-type control with odor and n = 85 for PDM-DN-silenced larvae. For more information about the statistics, see *Supplementary file 1*.

DOI: https://doi.org/10.7554/eLife.38740.002

The following figure supplements are available for figure 1:

**Figure supplement 1.** Technical drawing of the behavioral assay used for the loss-of-function behavioral screen.
DOI: https://doi.org/10.7554/eLife.38740.003

**Figure supplement 2.** Overview of the results of the loss-of-function screen.
DOI: https://doi.org/10.7554/eLife.38740.004

**Figure supplement 3.** Impairing PDM-DN function affects run-to-turn transitions without inducing motor deficits.
DOI: https://doi.org/10.7554/eLife.38740.005

located in segmentally-organized network elements in the VNC (*Fox et al., 2006*; *Inada et al., 2011*; *Berni et al., 2012*; *Berni, 2015*; *Lemon et al., 2015*; *Pulver et al., 2015*). Neural mechanisms that modulate coordinated rhythmic motor patterns have been uncovered in the VNC of the larva (*Kohsaka et al., 2014*; *Heckscher et al., 2015*; *Clark et al., 2016*; *Fushiki et al., 2016*; *Hasegawa et al., 2016*; *Yoshikawa et al., 2016*; *Matsunaga et al., 2017*). Mathematical modeling has tested the scope of mechanistic hypotheses related to the biomechanics of locomotion and the nature of proprioceptive feedback mediating body-environment interactions during exploratory behavior (*Gjorgjieva et al., 2013*; *Pehlevan et al., 2016*). The consistency of our present understanding of the sensorimotor mechanisms directing chemotaxis has been tested in computational models with increasing complexity and realism (*Davies et al., 2015*; *Wystrach et al., 2016*). In stark contrast with these advances, the neural substrates of sensory processing and action selection are still poorly characterized.

Descending neural pathways create a bottleneck in sensorimotor transformations: they create an interface between the circuits that carry out sensory processing in the brain lobes and those in charge of motor pattern generation in the VNC. Descending neurons are often regarded as key elements of 'command' circuits that condition the release of naturally occurring motor programs (*Kristan, 2008*). The leech provides a prime example of descending 'command' neurons with somas in the subesophageal zone (SEZ) where mechanosensory inputs are integrated to trigger segmental swim-initiating circuits in the VNC (*Brodfuehrer and Friesen, 1986*). Anatomical and physiological studies in other insects such as crickets (*Staudacher, 2001*), moths (*Kanzaki et al., 1994*), locusts (*Träger and Homberg, 2011*) and cockroaches (*Burdohan and Comer, 1996*) have also highlighted the regulatory importance of descending commands in the control of walking and flying behaviors. In adult *Drosophila*, activation of distinct descending neurons can elicit stereotypic behaviors such as backward walking (*Bidaye et al., 2014*) and courtship song (*von Philipsborn et al., 2011*). Recently, a detailed analysis of a large collection of Split-Gal4 driver lines targeting distinct descending neuron populations has concluded that most descending neurons produce a single behavioral output, while a minority of descending neurons are sufficient to trigger multiple behavioral outputs that are manifested either simultaneously or sequentially (*Cande et al., 2017*). In the larva, descending inputs from the brain lobes (BL) and SEZ are necessary to regulate the speed and frequency of the CPGs in charge of the actuation of forward runs, stop-turns and backward runs (*Berni et al., 2012*; *Pulver et al., 2015*; *Tastekin et al., 2015*).

Here, we identified a descending neuron (PDM-DN) whose artificial activation triggers run-to-turn transitions during *Drosophila* larval chemotaxis. Using electron microscopy (EM) reconstruction of its synaptic partners, we report that PDM-DN receives olfactory input from a set of lateral horn interneurons and that it provides outputs to a set of interneurons in the SEZ. We showed that PDM-DN blocks forward peristalsis by activating an inhibitory descending neuron in the SEZ, SEZ-DN1. By monitoring the propagation of waves of segmental contractions and motor neuron activity, we demonstrated that both PDM-DN and SEZ-DN1 cease forward locomotion by inhibiting the initiation of

new peristaltic waves at the posterior segments. Using EM reconstruction, we described that SEZ-DN1 synapses onto a network of excitatory premotor neuron A27h (*Fushiki et al., 2016*), which is thought to be important for forward peristaltic wave propagation. Strikingly, the connections of SEZ-DN1 onto the A27h network are restricted to the posterior segments, suggesting that SEZ-DN1 inhibits A27h neurons at the posterior segments to block the early phase of forward wave propagation. The analysis of the sensorimotor pathway described here paves the way to study how neural circuits underlying action selection are organized from the sensory neurons to the motor neurons.

## Results

### Identification of a descending neuron involved in the control of chemotaxis

To identify neurons in the larval brain involved in chemotaxis behavior, we conducted a loss-of-function (LoF) screen on a large collection of Split-Gal4 driver lines (*Li et al., 2014*; *Dionne et al., 2018*). Acute LoF impairments were achieved by blocking synaptic transmission with tetanus toxin light chain (TNT) (*Sweeney et al., 1995*). The screen was based on a modified version of a high-throughput olfactory assay introduced in previous work (*Tastekin et al., 2015*). The spatio-temporal resolution of the loss-of-function characterization was enhanced by using the multi-worm tracker (MWT) (*Swierczek et al., 2011*), which allowed us to track orientation behavior of multiple larvae simultaneously in a single trial (*Figure 1B*). The assay consisted of a large square dish coated with agar. The lid of the dish was outfitted with four hanging droplets of an attractive odor (*Figure 1—figure supplement 1*). This assay produced stable and reproducible odor landscapes (*Louis et al., 2008*). Each of the four odor sources created a gradient, which acted like a spatial attractor on the behavior of single larvae (*Figure 1Biii* and *Figure 1Biv*). We placed 20 larvae in the middle of the arena and tracked their trajectories for 5 min (*Figure 1Bi* and *Figure 1Bii*). We observed empirically that the initial orientation of a larva was a key factor determining the source under which it resided (*Figure 1Biii* and *Figure 1Biv*). Multiple sources were used to minimize interactions between individual larvae by distributing their accumulation across four regions of the plate (*Figure 1Biii* and *Figure 1Biv*).

Using Split-Gal4 driver lines, we achieved a more specific phenotypic characterization of the functional impairments induced by small neuronal subsets (*Pfeiffer et al., 2010*; *Dionne et al., 2018*). We screened ~300 Split-Gal4 driver lines most of which labeled only a single neuron per brain hemisphere (*Figure 1—figure supplement 2*). Silencing neurons labeled by eighteen different Split-Gal4 driver lines led to a statistically significant increase or decrease in chemotactic performances (*Figure 1—figure supplement 2*). This approach permitted the identification of a Split-Gal4 line that specifically labels an olfactory descending neuron (PDM-DN) connecting the lateral horn region of the brain lobes to the ventral nerve cord (*Figure 1C* and *Figure 2A*). The LoF phenotype associated with the expression of TNT in PDM-DN (PDM-DN>TNT) demonstrated an overall reduction in the accuracy of chemotaxis manifested as a wider spread of individual trajectories around the odor sources (*Figure 1D*). This chemotactic defect was quantified through the time course of the distance separating each animal from its closest odor source (*Figure 1E–F*). During the first three minutes of a trial, PDM-DN>TNT larvae (*Figure 1E*, magenta trace) remained further away from the source than their positive controls (*Figure 1E*, black trace). While parental-control larvae stayed at an average distance of ~13 mm to the nearest source between 120 and 240 s, PDM-DN silenced larva remained at a larger distance (~19 mm) during the same period of time. The LoF defect did not result from an overall decrease in locomotor speed (*Figure 1—figure supplement 3A*), but a significant decrease in turn rate (*Figure 1—figure supplement 3B*). In spite of this significant turning defect, the ability of PDM-DN >TNT larvae to orient toward the odor gradient was sufficient to produce significant accumulation under the source compared to the no-odor control (*Figure 1Biv and E and F*).

The PDM-DN neurons form a pair of bilaterally symmetrical cells in each brain lobe (*Figure 2Ai*). By using the MultiColor FlpOut (MCFO) technique to reveal the anatomy of individual neurons (*Nern et al., 2015*), we showed that each PDM-DN sends its dendrite and axon contralateral to its cell body (*Figure 2Aiv*). The soma of PDM-DN is located in the posterior-dorsal-medial region of the brain (*Figure 2Aii*). Closer inspection revealed that PDM-DN has dendritic arborizations around the mushroom body (MB) peduncle (the axonal bundle of Kenyon cells preceding the formation of the

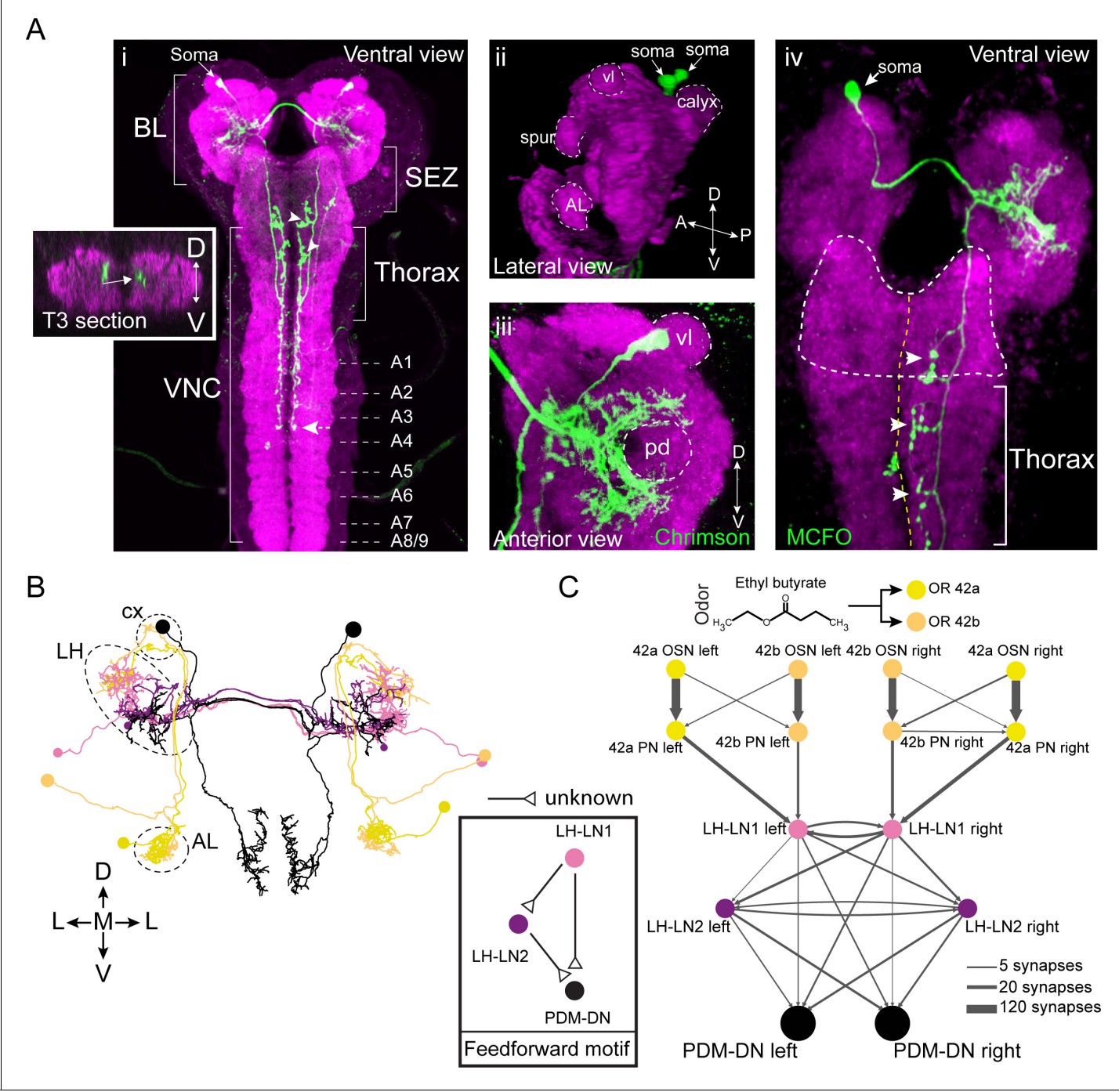

**Figure 2.** PDM-DN is a descending neuron that receives olfactory input. (**A**) Anatomical characterization of the PDM-DN neuron using light microscopy. (**Ai**) Anatomy of the PDM-DN neuron revealed by maximum projection of confocal sections. The soma of the descending neuron (PDM-DN) is located in the brain lobes (BL, arrow). Arrowheads indicate large axonal varicosities in the subesophageal zone (SEZ) and thoracic regions. The axon of the descending neuron extends posteriorly to the fourth abdominal segment (dashed arrow). Cross section of the third thoracic segment is shown in the inset. Arrow in the inset indicates the dorso-medial location of the PDM-DN axon. D: dorsal, V: ventral. VNC: ventral nerve cord, A1-A9 abdominal segments 1–9 of the VNC. (**Aii**) 3D reconstruction of the PDM-DN neuron (green) and larval neuropil (magenta). Antibody staining against CsChrimson:: mVenus expressed in PDM-DN neuron reveals that its soma is located in the posterior-dorsal-medial region of the brain lobes (arrows). The following brain regions are demarcated by dashed lines: antennal lobe (AL), tip of the ventral lobe (vl) of the mushroom body, spur and the mushroom body calyx. (**Aiii**) Dendritic arborizations of the PDM-DN neuron (green) are concentrated around the mushroom body peduncle (pd) and its surroundings. (**Aiv**) Use of the MCFO technique (**Nern et al., 2015**) to label a single PDM-DN neuron. The dendritic tree and the axon of the PDM-DN neuron are contralateral to its soma (arrow). Arrowheads indicate the large axonal varicosities in the anterior part of the PDM-DN axon. Green: PDM-DN neuron;
*Figure 2 continued on next page*

*Figure 2 continued*

magenta: neuropile. (B) EM reconstruction of the postsynaptic partners of PDM-DN. Anterior view of the *Or42a* and *Or42b* PNs (yellow and light pink); lateral-horn local interneurons (LH-LNs) upstream of PDM-DN (dark and light purple) and PDM-DN neuron (black). Antennal lobe (AL), lateral horn (LH) and mushroom body calyx (cx) are demarcated by dashed lines. Note that the PDM-DN neuron dendrites overlap with LH-LNs exclusively in the lateral horn region. (C) Reconstruction of the upstream partners of the PDM-DN neuron reveals that PDM-DN receives olfactory input via the lateral horn. The *Or42a* and *Or42b* OSNs are the main sensory neurons that are responsible for the attraction toward ETB, which is the odor used in the loss-of-function screen (*Figure 1*). PDM-DN receives input from the *Or42a* and *Or42b* projection neurons (PNs) via two local interneurons located in the lateral horn (LH-LN1 and LH-LN2), which form a feedforward motif (left inset). The signs of the interactions of the feedforward motif are unknown. Abbreviations used in the anatomical maps: A: anterior, D: dorsal, P: posterior, V: ventral.

DOI: https://doi.org/10.7554/eLife.38740.006

The following figure supplements are available for figure 2:

**Figure supplement 1.** EM reconstruction of the PDM-DN neuron and its main upstream partners.

DOI: https://doi.org/10.7554/eLife.38740.007

**Figure supplement 2.** A subset of olfactory sensory neurons target PDM-DN through LH-LN1.

DOI: https://doi.org/10.7554/eLife.38740.008

MB lobes) and near the lateral horn (LH) region, suggesting that PDM-DN receives olfactory input (*Figure 2Aiii*). The axon of PDM-DN extends to the VNC dorso-medially (*Figure 2Ai*, left inset) and targets three regions: the subesophageal zone (SEZ), the thoracic neuromeres and the first four abdominal neuromeres (*Figure 2Ai*). To identify the upstream partners of PDM-DN, we turned to the electron microscopy reconstruction of the whole central nervous system (CNS) of a 1st instar larva (*Schneider-Mizell et al., 2016*). We identified PDM-DN in the EM stack based on the unequivocal resemblance of the candidate neuron reconstructed from the EM and light microscopy (*Figure 2B* and *Figure 2—figure supplement 1A*). The morphological matching was based on the posterior-dorsal-medial location of the soma of PDM-DN, the dendritic branches in the region surrounding the peduncle of the mushroom bodies, and the characteristic medial projections of the output varicosities from the descending axon (*Figure 2—figure supplement 1A*).

To design the screen that led to the identification of PDM-DN (*Figure 1*), we selected ethyl butyrate (ETB), an odor that elicits robust and reproducible attractive response in wild-type larvae (*Asahina et al., 2009*). This odor predominantly activates two olfactory sensory neurons (OSNs) at the moderate concentrations used in our LoF experiments: *Or42a* and *Or42b* (*Kreher et al., 2008*). Since silencing PDM-DN produced a defect in chemotaxis to ETB (*Figure 1*), we hypothesized that PDM-DN must receive inputs from the *Or42a* and/or *Or42b* OSNs. We traced the pre-synaptic partners of PDM-DN all the way up to the sensory periphery and found that PDM-DN receives the strongest olfactory inputs from the *Or42a* and *Or42b* OSNs through the lateral horn region (*Figure 2C* and *Figure 2—figure supplement 2*). The uniglomerular projection neurons (PNs) downstream from the *Or42a* and *Or42b* OSNs directly innervate the local interneuron LH-LN1 in the lateral horn region (*Figure 2B–C* and *Figure 2—figure supplement 1B*). While LH-LN1 receives ipsilateral input from olfactory projection neurons, it sends axonal projections that bilaterally synapse onto PDM-DN and onto a second local interneuron in the lateral horn region, LH-LN2 (*Figure 2B–C*). In addition, LH-LN2 gives bilateral inputs to PDM-DN (*Figure 2C* and *Figure 2—figure supplement 1C*). The circuit motif encompassing LH-LN1, LH-LN2 and PDM-DN constitute a feed-forward motif (*Alon, 2006*) downstream from the olfactory projection neurons.

## PDM-DN participates in the sensorimotor control of reorientation maneuvers during larval chemotaxis

The defect in chemotaxis that follows impairment in the function of PDM-DN could result from an impairment in either/both essential sensorimotor tasks (*Gomez-Marin and Louis, 2012*): (i) triggering the onset of a turn when the larva moves down the gradient (*Figure 1Ai*) and/or (ii) directing turns toward the local odor gradient. The behavioral analysis presented in *Figure 1* did not allow us to discriminate between these two alternatives. We therefore re-analyzed the behavioral defects that resulted from a functional impairment of PDM-DN in conditions where a detailed correlation could be established between the sensory inputs and the behavioral outputs. To this end, we benchmarked the behavior of single larvae in an assay where a stable odor gradient was created based on a single source of ethyl butyrate (*Louis et al., 2008*).

Using infrared spectroscopy, the stability and geometry of the gradient was established as described in earlier work (*Gomez-Marin et al., 2011*; *Tastekin et al., 2015*). Consistent with the results of group behavior tested in the four-odor-source assay (*Figure 1D*), a visual inspection of the trajectories of PDM-DN>TNT and its parental controls revealed that silencing the function of PDM-DN compromised the ability of larvae to accurately locate the nearest odor source and to stay under this position (*Figure 3A–B*). By quantifying the orientation behavior through the distance to the source, we observed that both parental controls and the PDM-DN>TNT larvae accumulate near the source after a transient phase (~30 s) during which larvae moved up the gradient. Unlike the results of the four-odor-source assay (*Figure 1*), the average distance to the source of the PDM-DN silenced larvae gradually increased after ~120 s (*Figure 3C*, magenta trace) while it remained stationary for both parental controls during ~240 s (*Figure 3C*, blue and black traces). This increase in the distance to source might either reflect the cumulated effects of imprecisions affecting elementary reorientation maneuvers or a premature loss of interest in the odor source. While PDM-DN >TNT larvae were clearly capable of approaching the odor source at the onset of the experiment, their average distance to the source remained larger than those of both parental controls (*Figure 3C*). This difference is unlikely to be explained by an overall reduction in locomotor activity given that silencing PDM-DN did not significantly decrease the run speed of PDM-DN>TNT larvae compared to the slowest parental control (*Figure 3—figure supplement 1A and H*). We nonetheless observed significant differences in the run speeds of the two parental controls — an effect likely caused by genetic background. The apparent loss of interest of wild-type larvae in the odor after >240 s might be due to a gradual flattening of the odor gradient over time. Differences between the temporal profiles of the distance to the odor source between the single-odor-source assay (*Figure 3*) and four-odor-source assay (*Figure 1*) could result from the unique shapes of the odor gradients in both assays.

To characterize the nature of the defect induced by silencing PDM-DN, we quantified the turn rate in a time window (120–180 s) during the middle of the experiment. We selected this time window since the distance of PDM-DN-silenced larvae to the odor source increased during this period while the distance of the wild-type larvae remained stationary (*Figure 3C*, light green box). To make the turn rate independent of the variability in speed observed between the PDM-DN>TNT and one of its parental controls (*Figure 3—figure supplement 1A*), we measured the rate of turning per distance unit (cm). In agreement with the results of the four-odor-source assay (*Figure 1—figure supplement 2*), we found that PDM-DN>TNT larvae turned less frequently than either parental control (*Figure 3D*, left). The same conclusion held when turn rates were quantified based on the turns measured per time unit (min) (*Figure 3—figure supplement 1B*). We then conditioned the turn-rate analysis on the sensory experience of the larva: turn rates were measured separately for up-gradient and down-gradient runs (*Figure 3—figure supplement 1C*). A difference in turn rates was only observed for down-gradient runs. Therefore, we conclude that silencing PDM-DN reduces the likelihood that larvae terminate runs associated with negative changes in odor concentration. By contrast, the likelihood of turning during the up-gradient phase of a run was unaffected by the LoF manipulation. In addition, silencing of PDM-DN did not lead to a decrease in turn rate in the absence of odor gradients, supporting the idea that the PDM-DN LoF phenotype is not due to an overall decrease in run-to-turn transitions (*Figure 3—figure supplement 1G and H*). Again restricting our analysis to a fixed time window (120–180 s), we inspected the second sensorimotor task associated with larval chemotaxis: the control of the direction of turning. We calculated the probability of turning toward the direction of the local odor gradient. Larvae with silenced PDM-DN displayed no decrease in the accuracy of the orientation of their turns. Interestingly the parental control (SS01994 x $w^{-/-}$) showed a significant reduction in turning performances compared of PDM-DN>TNT and the other parental control (*Figure 3D*, right panel). This decrease in reorientation performances might be due to the higher locomotor speed displayed by this parental control (*Figure 3—figure supplement 1A*), which might compromise the speed-accuracy tradeoff necessary to accurately sample and navigate odor space (*Chittka et al., 2009*).

## Acute activation of PDM-DN elicits stopping and initiate reorientation maneuvers

The behavioral defect ensuing from the functional impairment of PDM-DN suggested that this descending neuron participates in the control of the timing of reorientation maneuvers (run-to-turn transitions, *Figure 1Ai*). To test the ability of PDM-DN activation to trigger transitions from runs to

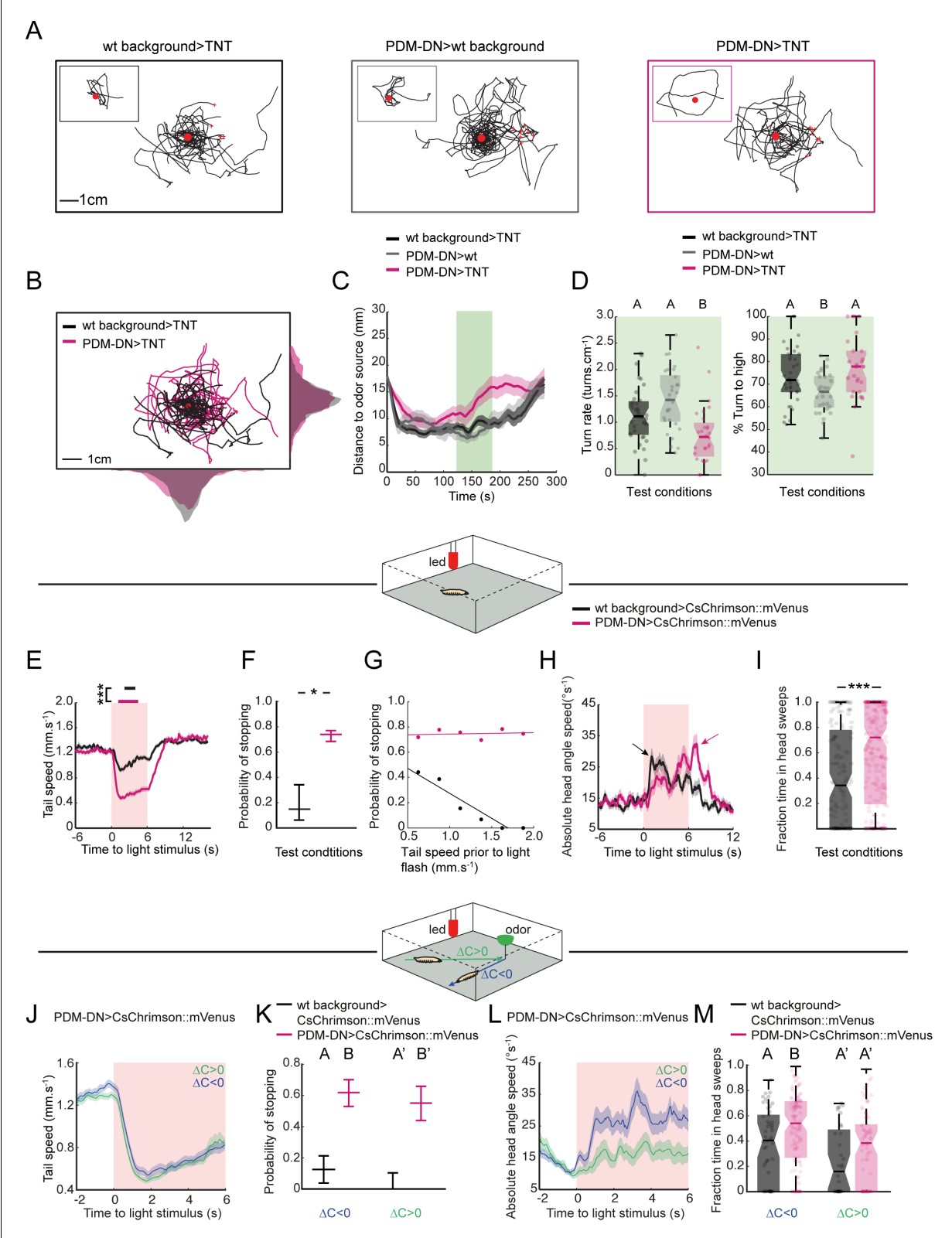

**Figure 3.** PDM-DN neuron plays a critical role in the sensorimotor control of reorientation behavior. (A–C): High-resolution analysis of the loss-of-function phenotype of the PDM-DN neuron in single larvae tracked in controlled odor gradients (*Tastekin et al., 2015*). (A) Set of 8 representative trajectories for both wild-type controls (left and middle panels, black and gray) and PDM-DN-silenced larvae (right panel, magenta). Individual representative trajectories are shown in the insets. Red points indicate the odor source and red crosses indicate the starting points of individual larvae.
*Figure 3 continued on next page*

*Figure 3 continued*

(B) The distribution of centroid positions along the x and y axes of the behavioral arena. Trajectories for wild-type controls (black) and PDM-DN silencing are superimposed (magenta). (C) Time course of the average distance of the centroids to the odor source. Shaded areas indicate SEM. PDM-DN-silenced larvae overshoot the odor source during 120–180 s (green box) while both wild-type controls stay around the odor droplet during this time window. (D) Left panel: comparison of turn rates. Right panel: percentage of turns directed toward higher odor concentration. Sample sizes: n = 36 larvae for wild-type control >TNT, n = 28 larvae for PDM-DN>TNT and n = 29 larvae for PDM-DN>wild type control. Different letters indicate statistically significant differences (Wilcoxon ranksum test. p<0.05 upon Bonferroni correction). (E) Optogenetic activation of the PDM-DN neuron evokes stopping behavior. Either larvae expressing CsChrimson::mVenus in the PDM-DN neuron (n = 35 larvae, 280 flashes) or wild-type controls (n = 20 larvae, 160 flashes) were exposed to 6 s red light flashes in the absence of an odor gradient. Acute optogenetic activation of the PDM-DN neuron induced a sharp decrease in tail speed leading to cessation of forward locomotion (see *Video 1*). Horizontal bars indicate the average durations of stops for wild-type controls (black) and PDM-DN activation (magenta). The bars are positioned according to the median latency of stops with respect to the flash onset. The average duration of stops for PDM-DN activation is significantly higher than that of wild-type controls (Wilcoxon ranksum test). For pairwise comparisons, the stars indicate a statistically significant difference between stop durations (***p<0.001). Shaded areas indicate SEM. (F) PDM-activated larvae show higher probability of stopping compared to the controls. The error bars indicate 95%-confidence intervals computed from binomial distributions (Clopper-Pearson method, p<0.05). The star indicates a statistically significant difference (*p<0.05). (G) The stopping behavior induced by the optogenetic activation of PDM-DN is not correlated with tail speed prior to the light flash ($R^2$ = 0.023, p=0.774). The low stopping probability due to the startle response in the wild-type controls is negatively correlated with the tail speed prior to the light flashes ($R^2$ = 0.903, p=0.0037). (H) Optogenetic activation of the PDM-DN neuron (same protocol as E) elicits an increase in head-angle speed (see head sweeps in *Video 1*). The increase in head-angle speed observed in wild-type larvae is due to a startle response induced at the onset of light flashes. The arrow indicates the maximum average head-angle speed reached right after the flash offset. (I) The fraction of time spent in head casting/turning mode during the light flash is significantly larger for the PDM-DN activated larvae compared to the wild-type controls (Wilcoxon ranksum test). Stars indicate a significant differences (***p<0.001). (J) Effect of optogenetic activation of PDM-DN in odor gradients. Same protocol as E except that a droplet of odor (100 mM ethyl butyrate, 5 µl) was suspended from the ceiling of the assay. The effect of PDM-DN activation was quantified for up-gradient and down-gradient runs. Acute activation of PDM-DN led to a strong decrease in tail speed irrespective of the direction of the runs. (K) Optogenetic activation of PDM-DN leads to a drastic increase in the probability of stopping during up-gradient and down-gradient runs. The error bars indicate 95% confidence intervals for binomial distributions (Clopper-Pearson method). (L) In PDM-DN activated larvae, head casts/turns are more strongly induced when the larva engages in a down-gradient run as shown in the time course of the head-angle speed. (M) The fraction of time spent in head sweeps is larger upon acute gain-of-function of the PDM-DN neuron. Different letters indicate statistically significant differences at p<0.05 (Wilcoxon ranksum test). For wild-type control, n = 50 flashes (down-gradient runs) and 32 flashes (up-gradient runs). For PDM-DN activation, n = 118 flashes (down-gradient runs) and n = 78 flashes (up-gradient runs). For more information about the statistics, see *Supplementary file 1*.

DOI: https://doi.org/10.7554/eLife.38740.009

The following figure supplements are available for figure 3:

**Figure supplement 1.** High-resolution analysis of the loss-of-function (LoF) phenotype of PDM-DN in two odor gradients.
DOI: https://doi.org/10.7554/eLife.38740.010
**Figure supplement 2.** Acute optogenetic activation of the PDM-DN neuron induces stopping behavior.
DOI: https://doi.org/10.7554/eLife.38740.011
**Figure supplement 3.** Acute optogenetic activation of the PDM-DN neuron induces stopping behavior.
DOI: https://doi.org/10.7554/eLife.38740.012
**Figure supplement 4.** Consistency of PDM-DN activation phenotype across individual larvae tested in absence of odor.
DOI: https://doi.org/10.7554/eLife.38740.013
**Figure supplement 5.** Consistency of PDM-DN activation phenotype across individual larvae tested in an odor gradient.
DOI: https://doi.org/10.7554/eLife.38740.014

turns, we acutely activated PDM-DN by using optogenetics. Upon expression of CsChrimson (*Klapoetke et al., 2014*) in PDM-DN, we subjected freely-moving larvae to 6-s-red-light flashes in a closed-loop tracking paradigm established previously (*Schulze et al., 2015*). Optogenetically-induced activation of PDM-DN produced a nearly deterministic switch from an ongoing run to a stop-turn (*Video 1*). The switch from running to stopping was highly reproducible (*Figure 3—figure supplement 2A*). A series of 8 consecutive light flashes produced eight transitions from a run (high-tail speed) to a stop-turn (low-tail speed) (*Figure 3—figure supplement 2A*). We adjusted the intensity of the red light to elicit stopping behavior in at least 75% of the flashes (*Figure 3—figure supplement 2B and D*) and focused on an intensity of 18 W/m$^2$ to minimize artefactual stops due to startle responses elicited by sudden exposure to intense red light (*Tastekin and Louis, 2017*). Averaging the time course of the tail-speed over 280 flashes yielded a stereotyped decrease in tail speed associated with the interruption of forward locomotion (*Figure 3E*). The non-zero value of the average tail speed resulted from inter-trial variability in the timing of stops (*Figure 3—figure supplement 4A*). By contrast, negative parental controls only showed a modest decrease in tail speed that is likely to arise from startle responses. The probability that wild-type controls stopped in response

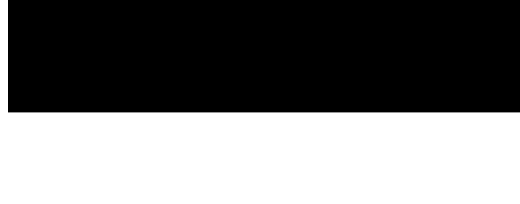

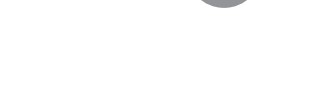

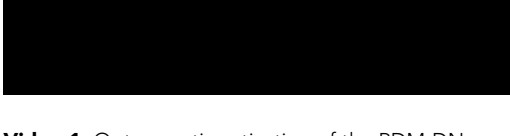

**Video 1.** Optogenetic activation of the PDM-DN neuron. Larva turns red when PDM-DN is activated. Head position is indicated by red dots and centroid position is indicated by a white line. The video sequence starts with the test condition (PDM-DN expressing CsChrimson) followed by the wild-type control (wild-type background crossed to UAS-CsChrimson).

DOI: https://doi.org/10.7554/eLife.38740.015

to light was nearly null and markedly lower than PDM-DN>CsChrimson larvae (*Figure 3F*). In few cases where wild-type larvae stopped, the average duration of stopping behavior was also shorter compared to the PDM-DN activation (*Figure 3E*). While we observed that the probability of stopping was anti-correlated with the tail speed in negative controls, no dependence between the stop probability and tail speed was found for PDM-DN>CsChrimson larvae, highlighting the deterministic nature of the optogenetic activation phenotype (*Figure 3G*). Finally, we examined whether prolonged activation of PDM-DN activity would maintain the larva in stopping mode for indefinite period of time. When subjected to a longer 20-s-red-light flash, the majority of PDM-DN>CsChrimson larvae remained in stopping mode for ~3 s, but the average tail speed gradually increased throughout the duration of the flash (*Figure 3—figure supplement 3A* and *Figure 3—figure supplement 4A*, right panels). This gradual increase in tail speed indicated a failure to sustain stopping behavior, thereby suggesting an adaptation of the PDM-DN neuron in response to persistent optogenetic activation or circuit-level mechanisms that bypass stopping behavior over longer durations (*Figure 3—figure supplement 4A*, right panels).

To define whether stops induced by optogenetic activation of PDM-DN activity are also followed by the initiation of a turning maneuver, we used the average head-angle speed to quantify head movements that precede turning upon PDM-DN activation. As a result of natural startle responses, parental negative controls displayed an increase in head-angle speed at the onset of the red-light flash (*Figure 3H*, black trace). Unlike the parental controls, the optogenetic activation of PDM-DN produced a delayed increase in the average head-angle speed peaking at ~3 s into the light activation (*Figure 3H*, magenta trace and *Figure 3—figure supplement 2C*). In addition, PDM-DN>CsChrimson larvae spent significantly more time sweeping their head throughout the light flash compared to the parental controls (*Figure 3I*). Unlike the decrease in tail speed, head-casting behavior induced by PDM-DN activation varied strongly across trials and animals (*Figure 3—figure supplement 4B and D*) and demonstrated a sensitivity to sensory experience (*Figure 3—figure supplement 5*). These results suggest that the effect of PDM-DN on the control of the head-casting dynamics is less direct than its inhibition of forward peristalsis. Next, we asked whether unilateral activation of the PDM-DN neuron was sufficient to trigger symmetrical inhibition of forward peristalsis or whether it resulted in asymmetrical inhibition leading to turns. To answer this question, we stochastically excised a stop cassette upstream of CsChrimson::mVenus coding sequence though a pan-neuronal expression of the low-activity flippase Flp2::PEST under the control of R57C10-Gal4. This way, we produced larvae with the expression of CsChrimson restricted to a PDM-DN neuron on one side of the brain (*Figure 3—figure supplement 3B*, left panel). Optogenetic unilateral activation of the PDM-DN neuron was sufficient to induce stopping and head-casting behaviors (*Figure 3—figure supplement 3B*, right panel).

In adult flies, the phenotypes elicited by gain-of-function manipulations of descending neurons could be influenced by the behavioral state of the animal (*Cande et al., 2017*). To determine whether the gain-of-function phenotype of PDM-DN was conditioned on the behavioral state of the larva, we exploited the closed-loop tracker to optogenetically activate PDM-DN when the larva was engaged in head casting behavior (*Figure 3—figure supplement 3C and D*). PDM-DN activation prolonged the ongoing stop (mean duration of stops: 4.66 ± 0.35 s, *Figure 3—figure supplement 3D*, magenta). When exposed to the same pattern of light stimulation, parental controls engaged in

head casting behavior switched to running after 1–2 head casts (mean duration of stops: 2.81 ± 0.11 s, *Figure 3—figure supplement 3C and D*, black). These results argue that PDM-DN-evoked stopping behavior does not depend on the behavioral state of the larva.

In the absence of olfactory sensory information, acute optogenetic activation of PDM-DN>CsChrimson larvae produced deterministic stops (*Figure 3E–F*, *Figure 3—figure supplement 2A* and *Figure 3—figure supplement 4A*). Building on this observation, we probed whether the release of stopping behavior was dependent on the sensory experience of the larva. To this end, we characterized the effects of optogenetic activation of PDM-DN during chemotaxis in a gradient resulting from a single-odor-source of ethyl butyrate (*Figure 3J–M*, top diagram). Light flashes were produced at random times and a *post-hoc* analysis enabled comparing the behavioral responses induced during up-gradient (toward the source) and down-gradient (away from the source) runs. While the sensory experience did not appear to affect the dynamics underlying switches from running to stopping modes elicited by optogenetic activation of PDM-DN (*Figure 3J–K* and *Figure 3—figure supplement 5A–B*), the head-casting behavior of PDM-DN->CsChrimson larvae showed a remarkable sensitivity to the sensory experience preceding a light flash: during down-gradient runs, larvae were more likely to engage in vigorous head casts than during up-gradient runs (*Figure 3—figure supplement 3F–G* and *Figure 3—figure supplement 5A–B*). Accordingly, the fraction of time spent casting was higher during down-gradient runs than during up-gradient runs (*Figure 3M*). Upon optogenetic activation, larvae that were moving down-gradient had a higher probability of implementing a head cast upon PDM-DN activation than larvae moving up-gradient (*Figure 3M* and *Figure 3—figure supplement 3G*, magenta). Negative controls responded to light flashes by engaging in head casts during both down-gradient and up-gradient runs (*Figure 3—figure supplement 3E and G*, black), suggesting that the startle response elicited by the light flashes override the olfactory information. In short, artificial activation of the activity of PDM-DN triggers deterministic pausing behavior while it can evoke turning behavior in an experience-dependent way.

## Activation of PDM-DN blocks the propagation of waves of forward peristalsis

As an initial step to determine the mechanisms of action of PDM-DN on the premotor system of the larva, we combined optogenetic activation of PDM-DN with a detailed kinematic analysis of the pattern of muscle contractions underlying locomotion in the larva. Forward locomotion is underlain by repetitive cycles of wave-like peristaltic contractions starting from the most posterior segments and ending at the anterior segments (*Heckscher et al., 2012*). As each of the eight abdominal segments sequentially contracts and relaxes from the most posterior to the most anterior segments, a peristaltic wave travels from the tail to the head. At the beginning of each peristaltic wave, the tail speed reaches its maximum due to a mechanism called visceral-piston phase (*Heckscher et al., 2012*). During the rest of the peristaltic cycle, the tail speed decreases before peaking again at the beginning of the next cycle. The periodic change in tail speed is illustrated in *Figure 4A* (bottom panel) for freely-moving larvae. When the wave terminates in the head (anterior) region, the body of the larva is fully extended and its length reaches its maximal amplitude (*Figure 4A*, top panel). Based on this phenomenology, the peristaltic wave cycle is defined as the period between a peak (maximum) in tail speed and the subsequent peak in body length. Given the cyclic nature of peristalsis, we described the state of the wave as an angle ranging from 0 (posterior end) to 360 (anterior end) degrees (*Figure 4A*).

Since the optogenetic activation (gain-of-function) of PDM-DN promotes stopping (*Figure 3E and F*), we hypothesized that the optogenetic activation of PDM-DN impaired the initiation and/or the propagation of forward peristaltic waves. In particular, we aimed to determine whether PDM-DN activation blocked (i) the propagation of ongoing waves and/or (ii) the initiation of future waves. To address this question, we coarsely described the peristaltic wave into three phase bins (*Figure 4A*, right panel): the initial phase covering the two most posterior abdominal segments (A7-A8, 0–90 degrees, blue); the middle or 'hinge' phase covering the middle segments (A4-A5, 135–225 degrees, green); the late phase covering the two most anterior abdominal segments (A1-A2, 270–360 degrees, red). Using this framework, we conducted a post-hoc analysis of the tail speed upon PDM-DN activation at different phases of the wave of peristaltic contraction. *Figure 4B* depicts the time course of the tail speed when the optogenetic activation of PDM-DN occurs during the initial (blue), the hinge (green) and the late (red) phases of peristalsis. In all cases, optogenetic activation of PDM-

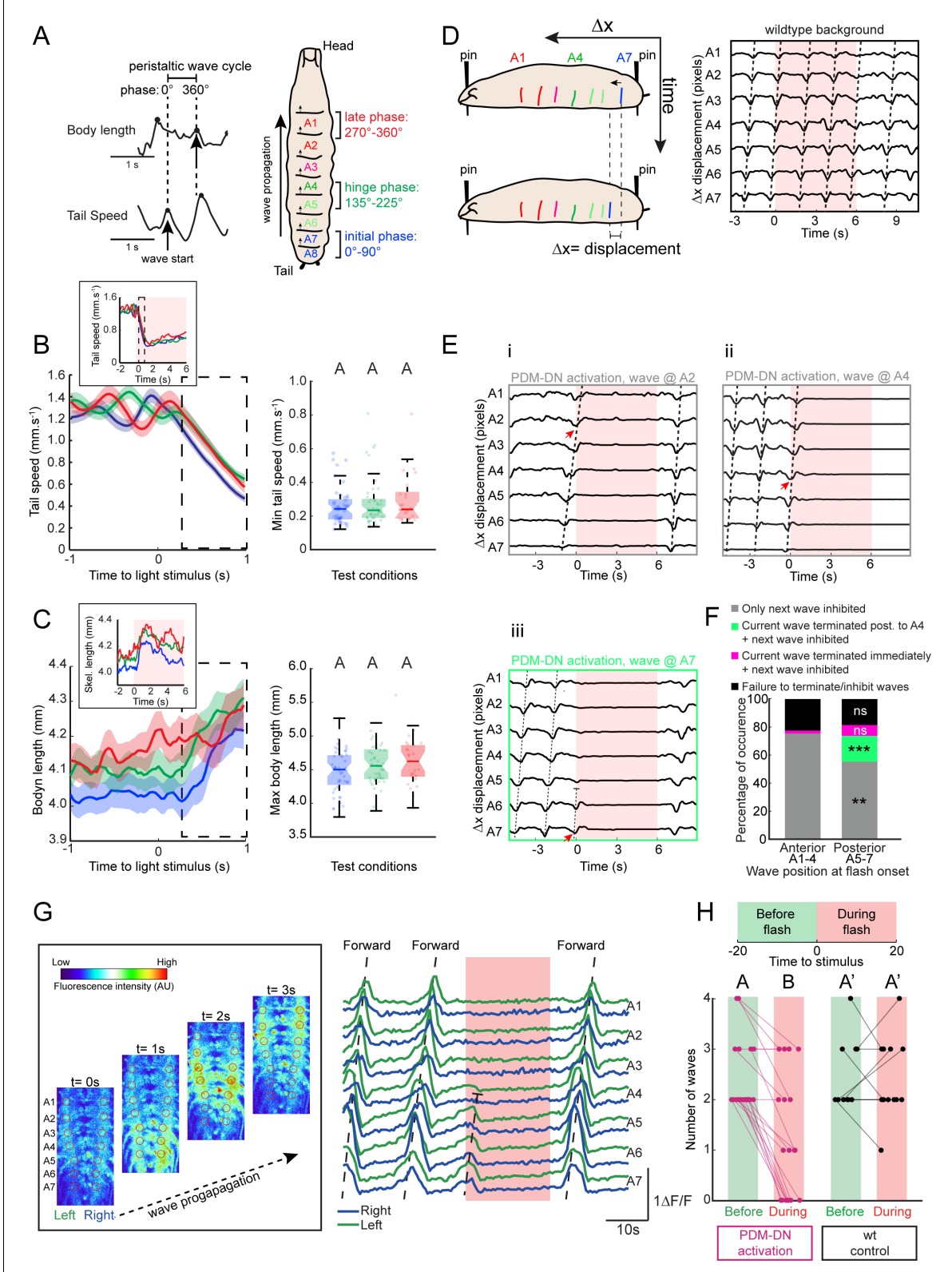

**Figure 4.** Detailed analysis of the effects of PDM-DN activation on segmental contractions and motor neuron activity. (**A**) Definition of forward-locomotion wave in freely moving larvae. During forward locomotion (run), tail speed and body length exhibit wave-like patterns. During the initiation of the forward wave, tail speed peaks due to the contraction of the most posterior segment (left panel, black arrow in tail speed). At the end of the wave of peristaltic contraction, larvae extend their body to its maximal length. This extension translates into a peak in the body length (left panel, black arrow

*Figure 4 continued on next page*

*Figure 4 continued*

in body length). (Left panel) We define the forward wave cycle as the time window between a peak in tail speed and the next peak in body length. (Right panel) Approximate mapping of phases of wave cycle on the abdominal segments (a full cycle spans 360 degrees). The initial phase of the wave corresponds to the contraction of the most posterior segments (initial phase, blue). As the wave is half-finished, segmental contractions reach the middle part of the body (green, hinge phase). The most anterior segments contract at the last phase of the wave (red, late phase). (**B–C**) Activation of the PDM-DN neuron at different phases of wave propagation. Optogenetic activation of the PDM-DN neuron during the initial, hinge and late phases. (**B**) PDM-DN activation evokes a drop in tail speed irrespective of the phase of the forward wave (left panel). The inset shows the mean tail speed throughout the flash. During the light flash (6 s), there is no significant difference in the minimum tail speed reached upon PDM-DN activation between the three phases considered in this analysis (right panel, Wilcoxon rank-sum test, p>0.05 upon Bonferroni correction). Each point in the background represents a single trial. Horizontal lines represent the median for each condition. Semi-transparent boxes represent the 25th and 75th percentiles. The whiskers extend to the most extreme data points excluding the outliers (default settings of the 'boxplot' function of Matlab_R2015b). Different letters indicate statistically significant differences. (**C**) Optogenetic activation of the PDM-DN neuron does not cease the wave propagation immediately. Upon light onset, the body length increases during the first second of the optogenetic activation of PDM-DN (dashed box to be compared to the dashed box of panel B). The inset shows the mean body length throughout the flash. During the light flash (6 s), there is no significant difference in the maximum body length reached upon PDM-DN activation between phases (Wilcoxon rank-sum test, p>0.05 upon Bonferroni correction). Each point represents a single trial. Horizontal lines represent the median for each condition. Semi-transparent boxes represent the 25th and 75th percentiles. The whiskers extend to the most extreme data points excluding the outliers (default settings of the 'boxplot' function of Matlab_R2015b). Different letters indicate statistically significant differences. Sample sizes: n = 45 trials for initial phase, n = 44 trials for hinge phase and n = 25 trial for the end phase. (**D–E**) Analysis of the PDM-DN activation phenotype at the level of segmental contractions using immobilized larvae. (**D**) To observe the effect of PDM-DN activation on the wave propagation at the level of segmental contractions, we immobilized the larva by pinning it down from both ends (tip of the head and the tail) on a transparent PDMS slab (left panel). Although the larva could generate forward peristaltic waves, it could not move forward, making it easier to track the segmental contractions (see *Video 2*). In this preparation, sequential contractions of the abdominal segments were clearly visible and quantified as displacement of each segment ($\Delta x$) along the anterior-posterior axis (right panel). (**E**) The effect of PDM-DN activity on the segmental contractions was observed by activating the PDM-DN neuron at different stages of the wave of peristaltic contraction. Prior to the light flash, larvae generated forward waves (tilted dashed lines) by sequentially contracting the abdominal segments starting at segment A7. The time course of segmental displacements is illustrated for conditions where the light onset coincides with contractions at segments A7 (Eiii), A4 (Eii) or A2 (Ei) (red arrows). (**F**) Different effects of PDM-DN activation on the ongoing peristaltic wave are reported as proportions of the total number of observations. For activation of PDM-DN while the wave was travelling through both the anterior and posterior segments, the ongoing wave reached and terminated at A1, but subsequent waves could not be initiated (gray). For posterior segments,~7% of the ongoing wave terminated immediately (purple) and a new wave could not be initiated throughout the flash. However no statistical difference is observed when compared to the anterior segments (Z-test, p=0.2076). About ~20% of the ongoing wave propagated up to the hinge point (segment A4) where they terminated (green color). In addition, new waves could not be initiated in this condition throughout the flash. This condition was never observed for the anterior segments (Z-test, p<0.05). The proportions of the failure to stop the wave is not significantly different for anterior and posterior segments (Z-test, p=0.6672). Sample size: 136 flashes (five preparations). Starts indicate statistically significant differences (**: p<0.01, ***p<0.001). (**G**) Calcium imaging of fictive locomotion patterns upon PDM-DN activation. We used isolated CNS preparations to analyze the effect of PDM-DN activation on fictive waves of forward locomotion. To record the motor neuron activity, we expressed the genetically-encoded calcium indicator GCaMP6f in glutamate-expressing neurons using the VGlut-LexA driver line. (Left panel) The picture sequence corresponds to snapshots at different time points during forward wave propagation (time interval between pictures: 1 s). Note the sequential increase in fluorescence intensity from A7 to A1. (Right panel) Prior to the light flash, larvae generated fictive forward waves (tilted dashed lines) manifested by sequential increase in the fluorescence levels starting from the posterior-most segment A7. The increase in fluorescence was symmetrically coordinated between the left and the right segments (left in green, right in blue). Upon optogenetic activation of the PDM-DN neuron, a fictive wave of locomotion starting from the posterior segments (**A7–A5**) terminated before reaching A4 (truncated dashed line). (**H**) To quantify the effects of the light-driven activity on PDM, we computed the number of forward waves before and during the light flash. While PDM-DN activation (magenta) leads to a significant decrease in the number of waves that were initiated and completed (n = 19 trials, Wilcoxon signed-rank test, p<0.05), wild-type controls (black) do not display a significant change (n = 12 trials). Each dot represents an independent trial. Different letters indicate statistically significant differences. For more information about the statistics, see *Supplementary file 1*.

DOI: https://doi.org/10.7554/eLife.38740.016

The following figure supplement is available for figure 4:

**Figure supplement 1.** Illustration of different effects of PDM-DN activation on ongoing waves of peristaltic contraction.
DOI: https://doi.org/10.7554/eLife.38740.017

DN was followed by a steep decrease in tail speed, indicative of stopping behavior. The minimum of the average tail speed was observed ~1 s after the onset of the light flash (*Figure 4B*). Similar results were observed for the body length (*Figure 4C*). Following optogenetic activation of PDM-DN, the average body length kept increasing steadily up to a maximum value independent of the phase at which the gain-of-function was implemented. Due to the gradual stretching of the larva during forward peristaltic waves, the body length started from a baseline value that was higher for the late phase than for the initial phase of the cycle. During the light flash, the larva remained fully extended for a couple of seconds before gradually relaxing (*Figure 4C*, top inset). Together, these observations reveal that PDM-DN activation does not result in an immediate freeze of peristalsis, but rather

that it allows the conclusion of the ongoing wave at the time of the flash and suppresses the initiation of subsequent waves.

To refine our analysis of the role of PDM-DN activation on the control of peristalsis, we tracked the position of individual body-wall segments, which can be visualized by the pigmented denticle bands on the ventral cuticle of the larva (*Berni, 2015*; *Pulver et al., 2015*). Larvae were restrained by locally immobilizing their head and tail with two micro dissection pins on a polydimethylsiloxane (PDMS) slab (*Figure 4D*, left panel and Materials and methods). While they were fixed in place, larvae still exhibited genuine patterns of peristaltic contraction (*Figure 4D*, right panel and *Video 2*). We monitored the movements of the denticle bands after optogenetically activating PDM-DN when the contraction wave coincided with each segment at least once. The sequential displacement of the denticle bands from A7 to A1 are reported in *Figure 4E*. When PDM-DN was activated during the late phase of the wave (denticle band displacement at segments A1-A3), the ongoing wave continued and terminated at the most anterior segments; subsequent waves were suppressed (*Figure 4Ei*). When PDM-DN activation coincided with the front of the wave activity at segment A4 or earlier, the wave either stopped immediately without propagating to anterior segments (*Figure 4—figure supplement 1A*) or it terminated at the most anterior segments (*Figure 4Eii*). In either case, subsequent waves were blocked. When the activation of PDM-DN coincided with the initial phase of the wave (denticle band displacement at segment A7), a striking phenotype was observed: the ongoing wave prematurely ceased before it reached the hinge point (denticle band displacement at segment A4) (*Figure 4Eiii*, but see *Figure 4—figure supplement 1B*) and the subsequent wave was also blocked. Finally, when optogenetic activation of PDM-DN coincided with the peristaltic wave at thoracic segments, subsequent waves could not be initiated while PDM-DN remained active (*Figure 4—figure supplement 1C–D*). Together, these observations suggest that artificial activation of PDM-DN is capable of terminating the peristaltic wave before but not after the hinge segment A4. Statistics about these different cases are reported in *Figure 4F*. While the results of the pinning experiments were consistent with the effect of PDM-DN activation on the tail speed and body length of freely moving larvae (*Figure 4A–C*), they also hinted at the potential existence of a different mode of action of PDM-DN on the anterior and posterior segments of the motor pattern generation in the VNC.

## Visualizing the effects of PDM-DN activation in the motor neurons during fictive locomotion

Motor neurons and abdominal muscles form a myotopic map: motor neuron dendrites are organized into distinct domains in the VNC mirrored by the segmental organization of their target muscles at the periphery (*Landgraf et al., 2003*). We turned to study the effects of PDM-DN activation on the motor neuron activity as a first step toward understanding the mechanisms of PDM-DN-induced pausing behavior. Isolated larval CNS devoid of sensory input can still produce rhythmic patterns of motor-neuron activity (fictive locomotion) with coordinated segmental activity similar to muscle contractions in intact larvae (*Lemon et al., 2015*; *Pulver et al., 2015*). Using a driver line specific to glutamatergic neurons to express a calcium indicator in motor neurons (CG9887-lexA > GCamP6f), we characterized the effects of optogenetic PDM-DN activation (PDM-DN>CsChrimson::mVenus) on fictive patterns of locomotion (*Figure 4G*). Periodic sequences of motor-neuron activity associated with forward locomotion were observed in isolated CNS preparations (*Figure 4G*, image sequence). Consistent with normal patterns of muscle contractions, motor-neuron activity started in the most posterior segments (A7) and sequentially progressed to the most anterior segment A1 (dashed line labeled as *F* in *Figure 4G*, right panel). In agreement with previous studies (*Berni, 2015*; *Pulver et al., 2015*), the frequency of wave generation was found to be ~5–10 times slower in isolated nervous system than during intact larval locomotion.

Using calcium imaging to monitor the rhythmic propagation of motor-neuron activity, we

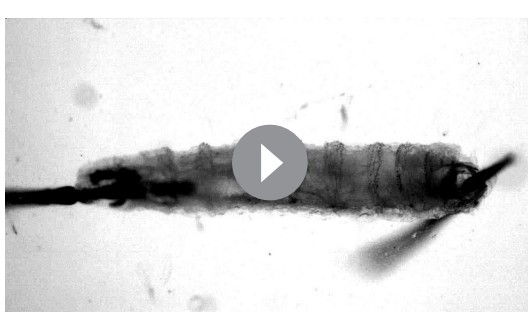

**Video 2.** Peristaltic wave propagation in a restricted larva.
DOI: https://doi.org/10.7554/eLife.38740.018

assessed the effect of optogenetic activation of the PDM-DN neuron. When the onset of the PDM-DN activation coincided with the beginning of a wave, the wave terminated at the hinge point (A4) (*Figure 4G*, right panel) — a result in agreement with the segmental contraction analysis (*Figure 4Eiii*). The initiation of subsequent waves was severely suppressed. To quantify the blocking effect of PDM-DN optogenetic activation on forward waves of motor activity, we quantified the number of complete waves observed 20 s before the onset of the light flash and 20 s through the light flash. We found that the number of waves strongly decreased after the flash compared to controls without optogenetic stimulation of PDM-DN (*Figure 4H*). By investigating the effects of PDM-DN activation on segmental contractions and motor-neuron activity, we concluded that PDM-DN is likely to induce pausing behavior by inhibiting the initiation of forward peristaltic waves in the posterior segments.

## Electron-microscopy reconstruction of the downstream partners of the PDM-DN neuron

The circuitry involved in the control of the propagation of forward peristaltic waves include a network of segmentally repeated excitatory premotor neurons called A27h (*Fushiki et al., 2016*). A27h excites motor neurons in the same segment, thereby promoting the contraction of longitudinal muscles involved in forward locomotion (*Kohsaka et al., 2012*). In addition, the A27h neurons of a given segment excite inhibitory GDL neurons in the adjacent anterior segment. In turn, the GDL neurons suppress the activity of the A27h neurons in the same segment. Therefore, A27h activity in a given segment leads to the contraction of the muscles in that segment while relaxing the muscles in the adjacent anterior segment by activating inhibitory GDL neurons. The connectivity pattern of the A27h neurons suggests that a mechanism based on the propagation of a wave of excitation-inhibition forms the pattern-generator-like mechanism underlying the propagation of forward waves of muscle contraction (*Pehlevan et al., 2016*). Since the activation of PDM-DN produces near-deterministic stops of forward locomotion (*Figure 3E–F* and *Video 1*), we speculated that PDM-DN must act on the activity of the A27h network. To determine the mode of action of PDM-DN on the premotor system, we turned to electron-microscopy (EM) to reconstruct the main pathway downstream from PDM-DN. EM connectivity data showed that PDM-DN is connected to different types of premotor neurons through a set of descending neurons in the SEZ region (*Figure 5A*, *Figure 5—figure supplements 1C* and *2C–D*).

EM reconstruction revealed that PDM-DN is connected to the A27h premotor network through a descending neuron named SEZ-DN1. SEZ-DN1 has its dendritic arbors in the SEZ region. This neuron projects its axon to the posterior abdominal segments of the VNC (*Figure 5C*, left and middle panels). As shown in the *Figure 5—figure supplement 1C*, SEZ-DN1 exclusively targets the A27h neurons located in the most posterior segments of the VNC. Remarkably, no synaptic connections are observed between the SEZ-DN1 and A27h neuron in the segments anterior to the hinge point defined by segment A4 (*Figure 4*). By screening the expression patterns of Gal4 driver lines (*Li et al., 2014*; *Dionne et al., 2018*), we identified a Gal4 line that specifically labeled the SEZ-DN1 neuron (R75C02, *Figure 5C* and *Figure 5—figure supplement 1D*). Given the suppressing effect of PDM-DN on forward peristaltic waves, we hypothesized that either PDM-DN or SEZ-DN1 must have an inhibitory effect on the activity of the A27h pre-motor neurons. To test this hypothesis, we profiled the neuro-transmitter(s) released by PDM-DN and SEZ-DN1. We performed immunostainings for the following three main neurotransmitters found in the larval brain: Gamma-aminobutyric acid (GABA), glutamate and acetylcholine.

Immunostaining for GABA and glutamate together with the labeling of PDM-DN (PDM-DN>CsChrimson::mVenus) showed that PDM-DN is unlikely to express GABA or glutamate (*Figure 5—figure supplement 1A*). By contrast, PDM-DN labeling showed co-localization with choline acetyltransferase (ChAT), the enzyme necessary for the synthesis of acetylcholine (*Figure 5B* and *Figure 5—figure supplement 1A*). This result suggested that PDM-DN was cholinergic. To obtain further evidence of the cholinergic nature of PDM-DN, we used RNA interference (RNAi) against ChaT while activating the PDM-DN neuron by using optogenetics. RNAi knock-down of ChaT abolished the PDM-DN gain-of-function phenotype (*Figure 5—figure supplement 1B*), indicating that acetylcholine release is necessary for PDM-DN-evoked pauses. The fact that PDM-DN is likely to be a cholinergic excitatory neuron implied that SEZ-DN1 must be inhibitory to block the activity of the A27h neuron. Accordingly, immunostaining for the inhibitory neurotransmitter GABA with the labeling of

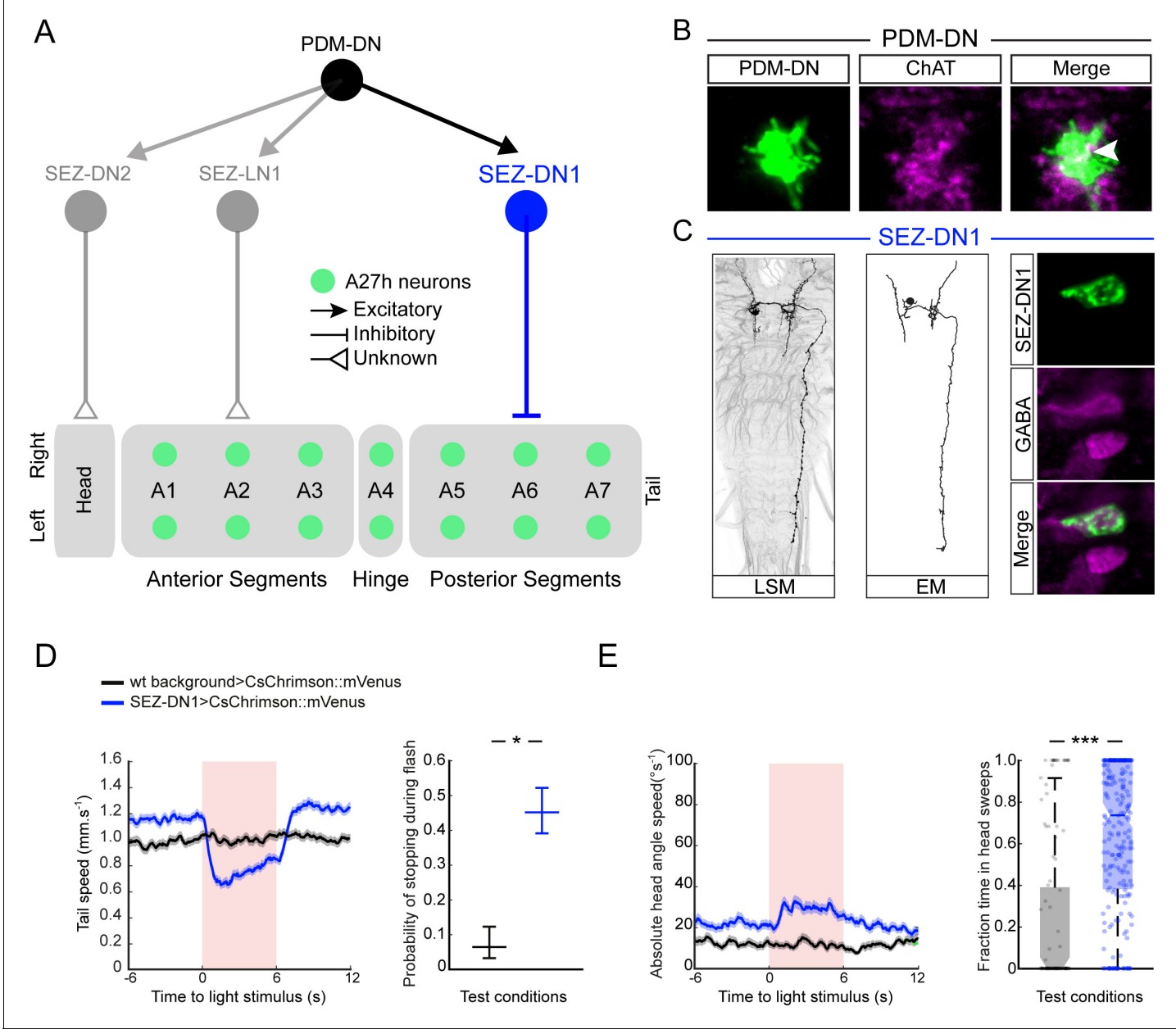

**Figure 5.** PDM-DN controls the premotor neurons involved in forward locomotion via a set of descending neurons located in the SEZ. (**A**) A set of neurons in the SEZ connects PDM-DN to the premotor system. PDM-DN is presynaptic mainly to three neurons: SEZ-LN1, SEZ-DN1 and SEZ-DN2. The SEZ-DN1 neuron gives inputs to segmentally repeated excitatory premotor neurons called A27h (see **Figure 5—figure supplement 1C**). The A27h targets of PDM-DN are exclusively in the posterior segments. The SEZ-DN2 neuron controls the prothoracic-accessory-nerve motor neurons (PaN motor neurons), which control head tilting (see **Figure 5—figure supplement 2C**). The SEZ-LN1 neuron gives input to premotor neurons mainly at the anterior segments (see **Figure 5—figure supplement 2D**). The nature of individual connections is reported according to the symbol shown in the top right panel. (**B**) The axonal varicosities of the PDM-DN neuron (green, antibody staining against CsChrimson::mVenus) co-localize with the immunostaining against choline acetyltransferase (magenta) suggesting that PDM-DN is a cholinergic excitatory neuron. (**C**) Identification of a driver line specific to SEZ-DN1. Light microscopy (left) and EM reconstruction (middle) of the SEZ-DN1 neuron. (Right panel) The soma of SEZ-DN1 (green) co-localizes with the immunostaining against GABA (magenta) suggesting that SEZ-DN1 is an inhibitory neuron. (**D**) Optogenetic activation of SEZ-DN1 is sufficient to elicit stopping behavior (see **Video 3**). Time course of the tail-speed (left panel) and quantification of the probability of stopping upon SEZ-DN1 activation (right panel). Stopping is significantly more likely to take place upon SEZ-DN1 activation than in wild-type controls. The error bars show 95% confidence intervals for binomial distributions (Clopper-Pearson method, p<0.05). Sample sizes: n = 120 trials for wild-type background and n = 248 trials for SEZ-DN1 activation. Star indicates statistically significant difference. (**E**) SEZ-DN1 activation leads to head casting/turning behavior. Time course of the head-angle speed (left panel) and fraction of time spent in head sweeps during the light flashes (right panel). SEZ-DN1 activation

*Figure 5 continued on next page*

*Figure 5 continued*

evokes significant head casting/turning behavior compared to the wild-type control (Wilcoxon ranksum test). Sample sizes: n = 120 trials for wild-type background and n = 248 trials for SEZ-DN1 activation. Stars indicate a statistically significant difference (***, p<0.001).

DOI: https://doi.org/10.7554/eLife.38740.019

The following figure supplements are available for figure 5:

**Figure supplement 1.** PDM-DN is an excitatory descending neuron that is connected to the premotor neurons located in the posterior segments via SEZ-DN1.

DOI: https://doi.org/10.7554/eLife.38740.020

**Figure supplement 2.** PDM-DN is strongly connected to a set of three SEZ neurons.

DOI: https://doi.org/10.7554/eLife.38740.021

SEZ-DN1 led to co-localization at the level of soma of this neuron (*Figure 5C*, right panel). Furthermore, optogenetic activation of the SEZ-DN1 neuron led to transitions from runs to stop/casts similar to PDM-DN (*Figure 5D–E*, *Video 3*). We conclude that SEZ-DN1 is a GABAergic neuron with an inhibitory effect on the A27h network of premotor neurons restricted to the posterior segments where forward waves of peristaltic contractions are initiated. In summary, PDM-DN was found to be differentially connected to the A27h neuron in the anterior and the posterior segments. In segments A6 and A7, the A27h neurons received bilateral inputs from the SEZ-DN1 descending neuron (*Figure 5—figure supplement 1C*), which itself received direct bilateral input from the PDM-DN neuron. Our finding that the A27h neurons located in segments anterior from A4 were unaffected by PDM-DN activity corroborated the absence of connectivity between SEZ-DN1 and A27 neurons in the abdominal segments anterior to A4.

PDM-DN has two additional post-synaptic partners in the SEZ that innervate known elements of the premotor system. First, SEZ-LN1 (*Figure 5A*, gray color and *Figure 5—figure supplement 2A and B*) is a local interneuron in the SEZ. SEZ-LN1 outputs on a premotor circuit that controls a set of motor neurons (RP2, RP3 and RP4) in segments T1, A1 and A2 (*Figure 5—figure supplement 2D*). Different RP neurons have been shown to control different sets of muscles (*Hoang and Chiba, 2001*; *Landgraf et al., 2003*; *Kim et al., 2009*). While RP2 motor neurons mainly control the dorsal muscles, RP3 and RP4 motor neurons connect to the ventral muscles (*Figure 5—figure supplement 2D*, inset). Interestingly, SEZ-LN1 connects contralaterally to RP2s and ipsilaterally to RP3 and RP4s. We speculate that asymmetrical contraction of dorsal and ventral muscles might contribute to the pattern of head sweeps elicited by optogenetic activation of PDM-DN. We identified a second descending neuron in the SEZ, named SEZ-DN2. This neuron directly innervates premotor neurons upstream from the motor neurons of the prothoracic accessory nerve (PaN) (*Figure 5—figure supplement 2A–C*) — a set of neurons that controls head tilting movements during feeding behavior (*Hückesfeld et al., 2015*). In particular, it is possible that the SEZ-DN2 neurons provide input to the PaN motor neurons, which execute head tilting maneuvers — a behavior that contributes to the head-scanning routine preceding the implementation of turns (*Green et al., 1983*; *Gomez-Marin et al., 2011*). As we could not identify Gal4-driver lines labeling the SEZ-LN1 and SEZ-DN2, these two pathways could not be studied further.

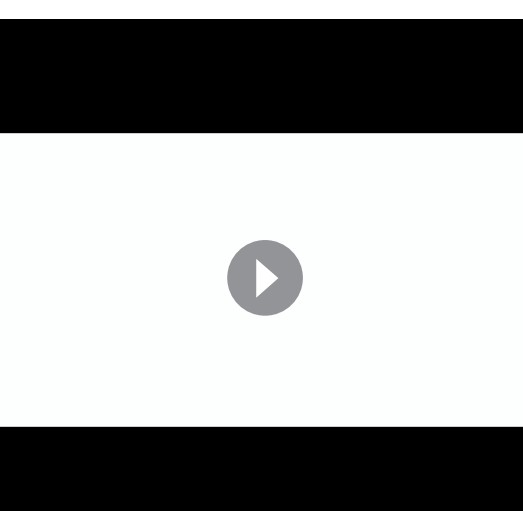

**Video 3.** Optogenetic activation of the SEZ-DN1 neuron. Larva turns red when SEZ-DN1 is activated. Head position is indicated by red dots and centroid position is indicated by a white line. The video sequence starts with the test condition (PDM-DN expressing CsChrimson) followed by the wild-type control (wild-type background crossed to UAS-CsChrimson).

DOI: https://doi.org/10.7554/eLife.38740.022

## Stopping behavior is elicited by blocking the initiation of peristaltic waves by SEZ-DN1

Using the same preparation as *Figure 4D–E* to monitor the propagation of peristaltic waves in fixed larvae, we analyzed the effect of SEZ-DN1 optogenetic activation on the premotor system. When SEZ-DN1 gain of function took place after the peristaltic wave had crossed the hinge point (segment A4), the wave tended to propagate up to the most anterior segment where it terminated. Subsequent waves were inhibited (*Figure 6A*). Consistent with the effects of gain-of-function manipulations of PDM-DN, SEZ-DN1 optogenetic activation was sufficient to block the propagation of forward wave when the wave was still located in the region posterior to the hinge point (*Figure 6Aiii and B*). Next, we corroborated these results by imaging the effects of SEZ-DN1 optogenetic activation on waves of motor neuron activity. Spontaneous forward waves of activity in motor neurons of the VNC were suppressed upon optogenetic activation of SEZ-DN1 (*Figure 6C*). As reported in *Figure 6D*, the activation of SEZ-DN1 drastically reduced the number of fictive waves of motor neuron activity observed before and during the light flash. Using calcium imaging, we corroborated the idea that PDM-DN has an excitatory effect on the activity of SEZ-DN1. By expressing CsChrimson in PDM-DN and imaging the activity of SEZ-DN1 with GCaMP6f, we observed that the activity of PDM-DN excited neural activity in the axons of SEZ-DN1 (*Figure 6E–F*). These experiments validate a model where PDM-DN activates SEZ-DN1, which in turn inhibits the activity of the A27h pre-motor system.

## Discussion

The *Drosophila melanogaster* larva has a numerically simple nervous system that comprises ~10,000 neurons (*Li et al., 2014*) directing a rich repertoire of behaviors that includes navigation in chemical, light and thermal gradients (*Almeida-Carvalho et al., 2017*). The larva displays stereotyped behavioral programs that can be decomposed into forward motion ('run'), locomotor pauses ('stops') followed by exploratory lateral-head movements ('head casts') and turns (*Green et al., 1983*; *Sawin-McCormack et al., 1995*; *Berni et al., 2012*; *Gomez-Marin and Louis, 2012*). Although the decomposition of the behavioral continuum displayed by the larva into discrete 'actions' represents an approximation (*Szigeti et al., 2015*), this approximation has proved valuable in various model organisms (*Berman et al., 2014*; *Wiltschko et al., 2015*), and it permitted the identification and functional characterization of neural circuits in *Drosophila* (*Vogelstein et al., 2014*; *Tastekin et al., 2015*; *Cande et al., 2017*; *Humberg et al., 2018*). The sensorimotor algorithm directing innate navigation in the larva is shared across sensory modalities (*Luo et al., 2010*; *Gomez-Marin et al., 2011*; *Gershow et al., 2012*; *Kane et al., 2013*). Movements toward favorable directions elongate runs, whereas movements toward unfavorable directions promote turning. The goal of the present study was to identify the neural circuits that implement the sensorimotor conversion of the OSN activity into the probability of switching from a run to a stop-turn.

Neural circuits in the brain are connected to the premotor system in the VNC by descending nerve fibers. In adult flies, descending neurons represent a relatively small population of ~1100 cells (*Hsu and Bhandawat, 2016*), accounting for less than 1% of the total number of neurons in the nervous system. By establishing the main connections between the centers carrying out sensory processing in the brain and the central-pattern-generating (CPG) circuits in the VNC, descending neurons are thought to play a key role in the control of sensorimotor behaviors. In adult flies, activation of the 'moonwalking' descending neuron induces backward locomotion (*Bidaye et al., 2014*). In larvae, activation of the recently-identified 'mooncrawler' neuron triggers backward locomotion and blocks forward locomotion (*Carreira-Rosario et al., 2018*). Complex sequences of actions can be elicited by the activation of a single descending neuron, such as courtship song production (*von Philipsborn et al., 2011*) and flight escapes (*von Reyn et al., 2014*; *von Reyn et al., 2017*). Using a collection of Split-Gal4 driver lines that labels relatively sparse sets of descending neurons (*Namiki et al., 2017*), a majority of descending neurons was found to elicit only one stereotyped behavior (*Cande et al., 2017*), but the same behavior could be elicited by distinct descending neurons. The behavioral effects of gain-of-function manipulations showed dependence on the ongoing motor state of the animal. By contrast, very little is known about the number and the organization of descending neurons in the larva. To identify descending neurons participating in the control of

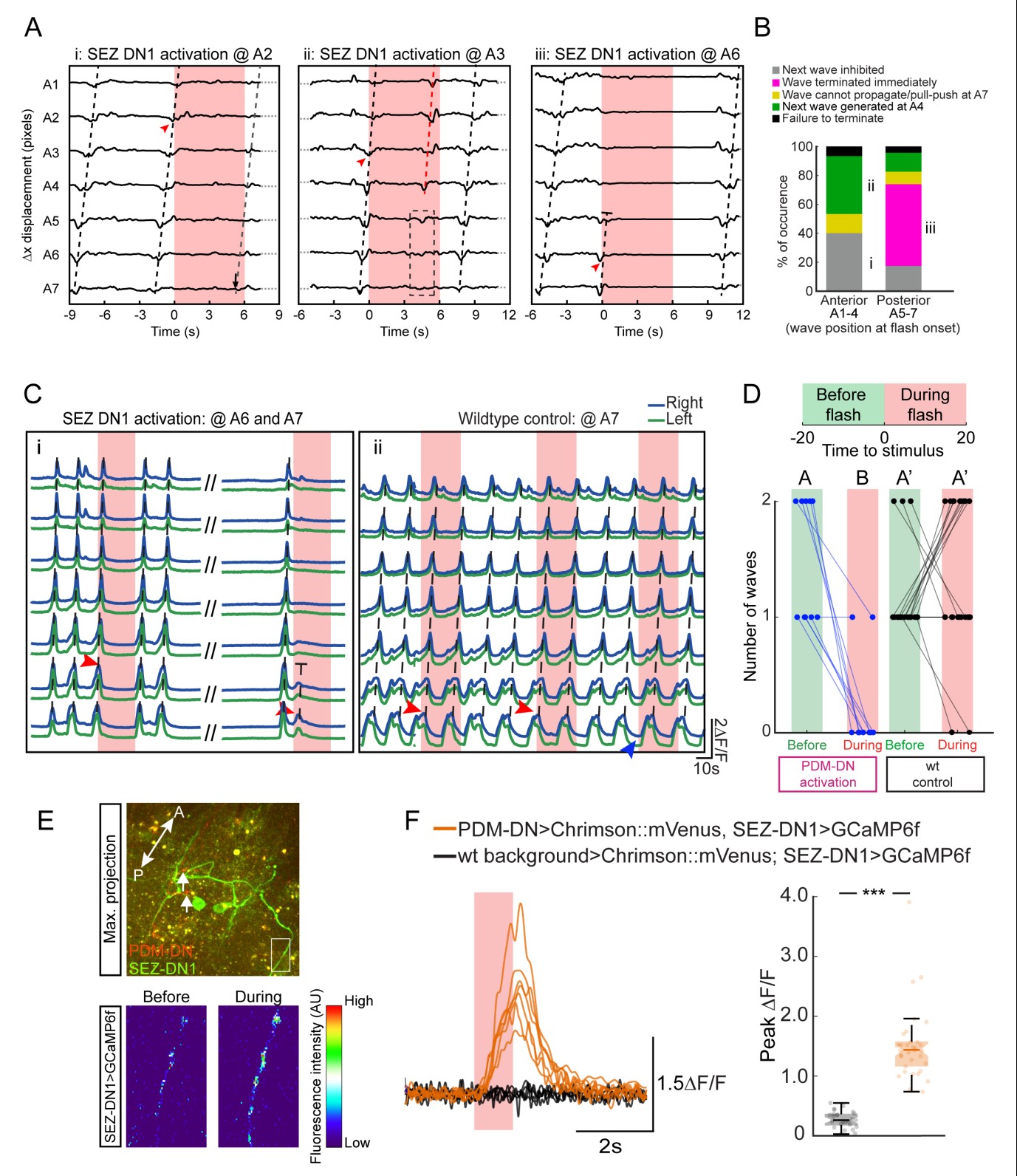

**Figure 6.** Detailed analysis of the gain-of-function phenotype of the SEZ-DN1 neuron at the level of segmental contractions and motor neuron activity. Activation of SEZ-DN1 inhibits the forward wave propagation at the posterior segments of the VNC. (A) Decomposition of the SEZ-DN1 activation phenotype at the level of segmental contractions using immobilized larvae. The analysis results from the same protocol as *Figure 4D*. SEZ-DN1 was optogenetically activated when forward wave was at A2 (left), A3 (middle) or A6 (right). (Ai) When SEZ-DN1 activation coincided with wave propagation

*Figure 6 continued on next page*

*Figure 6 continued*

at A2 (red arrowhead), the current wave was likely to terminate but the initiation of a new wave was blocked for the duration of the light flash. (**Aii**) Occasionally a forward wave was initiated from segment A4 to segment A1 during the optogenetic activation of SEZ-DN1 (red dashed line, middle panel). However, no contractile activity was observed at the level of the posterior segments (dashed rectangle). (**Aiii**) SEZ-DN1 activation ceased ongoing forward wave immediately at the posterior segments. In this example, activation of SEZ-DN1 when the wave was at segment A6 (red arrowhead) led to an immediate termination of the wave (truncated dashed line). (**B**) Main effects of SEZ-DN1 activation on the propagation of the peristaltic wave reported as proportions. If SEZ-DN1 was activated when the forward wave was at the posterior segments, the wave often terminated immediately (magenta, see also panel Aiii). In ~20% of the cases, the next wave was inhibited during SEZ-DN1 activation (gray). Occasionally, a new forward wave started at A4 (green). We also observed a pull-push behavior in which the posterior segments engage in a loop of contraction-relaxation without wave propagating to more anterior segments (yellow, data not shown). Activation of SEZ-DN1 when the wave had reached the anterior segments led to a suppression of the initiation of new waves in ~40% of the cases (gray). Forward waves initiated at A4 were observed more frequently upon optogenetic activation of SEZ-DN1 when the ongoing wave was located in the anterior segments (~40%, green). For both the anterior and posterior segments, a small proportion of the optogenetic gain-of-function failed to affect the forward wave propagation (black). (**C**) Imaging of fictive locomotion patterns upon SEZ DN1 activation. Same protocol as *Figure 4G* (Ci) SEZ-DN1 optogenetic activation when the fictive wave was at A6 (first flash, red arrowhead) or at A7 (second flash, red arrowhead). Following the first flash, the ongoing wave terminated and a new wave could not be initiated during the duration of the flash. In subsequent flashes, the motor neuron activity was suppressed immediately at A6-7 and the wave stopped before reaching A5 (truncated dashed line). The entire course between the two flashes is not shown for clarity (dashes). (**Cii**) A wild-type control larva was exposed to the same flashes of light as the SEZ-DN1 >Chrimson. The light flashes had no impairing effects on forward wave initiation/propagation such that new forward waves could be initiated and completed during the light flashes. (**D**) Quantification of the effect of SEZ-DN1 activation on fictive forward locomotion. Same data-quantification protocol as *Figure 4H*. Optogenetic activation of the SEZ-DN1 neuron led to a decrease in number of forward waves (Wilcoxon signed-rank test, $p<0.05$) while wild-type controls were not affected by light flashes (Wilcoxon signed-rank test, $p<0.05$). Sample sizes: n = 23 trials for wild-type controls and n = 10 trials for SEZ-DN1 activation. Different letters indicate statistically significant differences. (**E–F**) Calcium imaging of SEZ-DN1 activity upon optogenetic activation of PDM-DN. (E-top panel) The approximate region of interest (ROI) for the quantification of SEZ-DN1 activity is highlighted with a rectangle. Axonal varicosities of PDM-DN (red) are faintly visible (arrows). A: anterior, P: posterior. (E-Bottom panel) Pseudo-colored illustration of the pattern of activity in the ROI shown in the top panel, before (left) and during PDM-DN activation (right). (**F**) Quantification of the SEZ-DN1 activity upon PDM-DN activation. (Left panel) Each CNS preparation was exposed to the same light-flash protocol five times. Quantification of the mean ΔF/F for each preparation. Upon PDM-DN activation (semitransparent red box), SEZ-DN1 activity increased drastically (orange). By contrast, SEZ-DN1 did not respond to the light flashes in the absence of CsChrimson::mVenus in the PDM-DN neuron (black). (Right panel) Boxplot of the peak responses measured on each trial. PDM-DN activation led to significant increase in the activity of SEZ-DN1 (Wilcoxon ranksum test). The line centered on the 'waist' represent the median of each experimental condition. Semi-transparent boxes represent the 25th and 75th percentiles. The whiskers extend to the most extreme data points excluding the outliers (default settings of the 'boxplot' function of Matlab). Starts indicate statistically a significant difference (***$p<0.001$). Sample sizes: n = 25 trials (five preparations) for SEZ-DN1 in the absence of PDM-DN activation and n = 35 trials (seven preparations) in the presence of PDM-DN activation. For more information about the statistics, see *Supplementary file 1*.

DOI: https://doi.org/10.7554/eLife.38740.023

innate larval chemotaxis, we carried out a behavioral screen on a collection of sparse driver lines (*Li et al., 2014*; *Dionne et al., 2018*).

The behavioral screen was designed based on two assumptions. First, larvae display two types of navigational behaviors: attraction — the most common response elicited by volatile odors (*Cobb, 1999*; *Fishilevich et al., 2005*; *Mathew et al., 2013*) — and repulsion, a behavior elicited for chemical alarm cues such as the pheromone emitted by a natural predator of the *Drosophila* larva, the parasitoid wasp (*Ebrahim et al., 2015*). Based on the selectivity of the behavioral responses induced by individual descending neurons in adult flies (*Cande et al., 2017*), we reasoned that attractive and aversive responses might be controlled by different descending pathways. To focus on positive (attractive) chemotaxis, we devised an assay that elicited purely attractive behavior. Second, the functional deconstruction of the peripheral olfactory system of the larva has shown that single olfactory sensory neurons (OSNs) are sufficient to direct robust chemotaxis (*Fishilevich et al., 2005*; *Louis et al., 2008*; *Gershow et al., 2012*; *Hernandez-Nunez et al., 2015*; *Schulze et al., 2015*). We assumed that the activity of a single OSN, that expressing *Or42a*, was more likely to feed into a single descending neuron than the activity of an ensemble of OSNs, which might activate multiple descending pathways. For this reason, the screen was conducted with ethyl butyrate, an odor that primarily binds to the *Or42a* odorant receptors (*Kreher et al., 2008*).

Our loss-of-function screen led to the identification of two main classes of neuronal subsets with a phenotypic defect in innate chemotaxis (*Figure 1—figure supplement 2*). The first class labeled different sets of mushroom body (MB) neurons (i.e. Kenyon cells, mushroom body input neurons and mushroom body output neurons). Although the MB is not traditionally associated with the control of innate orientation behavior, recent work has uncovered that the MB participates in the control of

chemotaxis in adult flies (*Wang et al., 2003*; *Lewis et al., 2015*; *Perisse et al., 2016*; *Tsao et al., 2018*). Considering the effects of MB impairment on learned olfactory behaviors (*Owald and Waddell, 2015*), it is possible that a loss-of-function of particular subsets of MB neurons unbalances the net MB output, thereby producing a dysfunction in innate chemotaxis. The second class of neurons that our loss-of-function screen pointed out included descending neurons (*Figure 1—figure supplement 2A–B*). Given that descending neurons form a bottleneck in sensorimotor pathways, we concentrated on this neuron class in the rest of the work. Among the descending neurons identified in our behavioral screen, the anatomical features of the PDM-DN stood out as promising (*Figures 1C* and *2A*): the dendritic arborizations of this descending neuron cover regions of the lateral horn (LH) and the MB peduncle. On the output side, PDM-DN has large axonal varicosities in the subesophageal zone (SEZ) — a region previously implicated in the control of run-to-turn transitions during larval chemotaxis (*Tastekin et al., 2015*). The axon terminals of PDM-DN extend dorsally to the 4$^{th}$ abdominal segment, suggesting that this neuron might directly act on the premotor system (*Kohsaka et al., 2012*). Altogether, PDM-DN emerged as a strong descending-neuron candidate that transforms information about the larva's sensory experience collected from the LH and the MB into a modulation of the larva's motor output.

Using a set of complementary manipulations to test the effects of silencing or activating PDM-DN, we examined the role of this neuron on specific aspects of the sensorimotor control of innate chemotaxis. First, we demonstrated that PDM-DN activity contributes to the proper timing of run-to-turn transitions during chemotaxis. During down-gradient runs, the detection of negative changes in odor intensity leads to a graded increase in the probability of switching from a run to a turn (*Schulze et al., 2015*). Abrupt termination of the *Or42a*-OSN activity triggers near deterministic stops (*Schulze et al., 2015*). Upon constitutive loss-of-function of PDM-DN, larvae had a significantly lower probability of turning (*Figure 3D*). As a result of the inaccurate timing of their turns, larvae with impaired function of PDM-DN were unable to accumulate in the vicinity of the odor source with the same precision as their controls (*Figures 1D–E and* and *3B–C*). Remarkably, manipulations inducing a loss of function of PDM-DN did not affect the ability to turn toward the gradient (*Figure 3D*, right panel), arguing that distinct sensorimotor pathways control the timing and the direction of turning maneuvers. By expressing CsChrimson in the PDM-DN neuron, we established that acute optogenetic activation of PDM-DN elicits near deterministic stops (*Figure 3E–F*). Upon prolonged gain-of-function stimulations, the release of stopping behavior was accompanied by lateral head casts (*Figure 3H–I*). Together these results indicate that PDM-DN acts as a command-like element in the sensorimotor pathway that converts changes in the activity of the *Or42a* and *Or42b* OSNs into the probabilistic release of reorientation maneuvers. The primary effect of the activity of PDM-DN is to promote switching between run and stop-turn behaviors. Its secondary effect is to induce exploratory scans of the local odor gradient in preparation of a turn.

If PDM-DN is part of the sensorimotor pathway controlling chemotaxis, its activation must be dependent on the present —and potentially past— activity of the peripheral OSNs. We tested this hypothesis in functional perturbation experiments. We reported that the PDM-DN silencing phenotype depends on the olfactory sensory information: in odor gradients, silencing PDM-DN activity affected the release of turns during down-gradient runs, but not during up-gradient runs (*Figure 3—figure supplement 1C*). By contrast, silencing PDM-DN did not affect the basal turn rate in the absence of odor gradients (*Figure 3—figure supplement 1G and H*). We also tested the effects of the ongoing activity of peripheral OSNs on the release of stops upon medium-intensity optogenetic activation of PDM-DN (*Figure 3—figure supplement 2B*). Larvae carrying a PDM-DN>Chrimson transgene were optogenetically stimulated during up-gradient and down-gradient runs. Interestingly, no significant difference was found in the probability of releasing a stop-turn maneuver (*Figure 3K*). This trend was further confirmed by comparing the time course of the tail speed —a proxy for stopping behavior— before and during optogenetic stimulation for up-gradient and down-gradient runs. No difference was found between up-gradient and down-gradient runs (*Figure 3J*). A detailed inspection of the behavior associated with individual trials led to the same conclusion (*Figure 3—figure supplements 4A* and *5A*). This result suggests that PDM-DN itself does not integrate the history of the activity of *Or42a* and *Or42b* OSNs, otherwise the gain-of-function perturbations should have produced a higher probability of triggering stops during down-gradient runs compared to up-gradient runs. It is also possible that the light stimulation used in the experiments of *Figure 3J* was still too high to reveal differences due to the integration of distinct sensory experiences

between up-gradient and down-gradient runs. By comparison, we observed that the secondary effect of optogenetic activation of PDM-DN — the promotion of head casting— was strongly dependent on the ongoing olfactory experience of the larva: during down-gradient runs, PDM-DN activation led to vigorous and wide-amplitude head casts, whereas PDM-DN activation led to milder casting behavior during up-gradient runs (*Figure 3L*). Based on this result and the observation that the PDM-DN loss-of-function does not affect the accuracy of individual turns (*Figure 3D*, right panel), we propose that the head-casting component of reorientation maneuvers is gated by the activity of PDM-DN, but that it is also controlled by other descending pathway(s) that integrate the ongoing activity of the olfactory system.

The larval nervous system is amenable to a detailed reconstruction of neural circuits through electron microscopy (EM) (*Cardona et al., 2010*; *Ohyama et al., 2015*; *Schneider-Mizell et al., 2016*). While the EM reconstruction is achieved in the nervous systems of younger larvae (first-instar L1 developmental stage) than those tested behaviorally (third instar, L3), we have recently shown that the sensorimotor circuit involving the control of innate chemotaxis is fully functional at the L1 stage (*Almeida-Carvalho et al., 2017*). By comparing the anatomy of the PDM-DN neuron between light-scanning and EM microscopy, we pinpointed the PDM-DN neuron in the EM stack (*Figure 2* and *Figure 2—figure supplement 1A*). We reconstructed the main pre-synaptic partners of PDM-DN all the way to the peripheral olfactory system (*Figure 7A*). This reconstruction relied on earlier work in which we fully mapped the circuit diagram of the larval antennal lobe at the resolution of single synapses (*Berck et al., 2016*). PDM-DN receives olfactory inputs in the lateral horn region via two lateral-horn interneurons that form a feedforward circuit (*Figure 2B–C* and *Figure 2—figure supplement 1B–C*). Consistent with the fact that the loss-of-function screen involved an odor (ethyl butyrate) that only activates a small number of OSNs, we found that PDM-DN receives olfactory inputs from a subset of OSNs activated by this odor: *Or42a* and *Or42b* OSNs (*Figure 2C* and *Figure 2—figure supplement 2*). Given the incompleteness of the loss-of-function phenotype of PDM-DN (*Figure 3*), we speculate that redundant descending pathways controlled by the same set of OSNs might trigger different behavioral modules in a context-dependent manner.

What is the logic underlying the transformation of the activity patterns of OSNs into the all-or-none activation of the PDM-DN neuron? On the one hand, positive gradients promote sustained activity of the *Or42a* and *Or42b* OSNs (*Kreher et al., 2008*; *Schulze et al., 2015*). The *Or42a* OSN encodes the time derivative (slope) of ramps of odor concentrations (*Schulze et al., 2015*). On the other hand, the suppression of stop-turn during up-gradient runs implies that the activity of PDM-DN must be negatively correlated with the activity of the *Or42a* and *Or42b* OSNs. Strong activation of these two OSNs must suppress the activity of the PDM-DN neuron while inhibition of these OSNs must trigger the firing of PDM-DN (*Figure 7B–C*). Although the activity of the uniglomerular *Or42a* and *Or42b* PNs is expected to be roughly proportional to the activity of their cognate OSNs (*Asahina et al., 2009*), it is possible that these PNs extract higher-order features from dynamic patterns of odor concentrations, such as the acceleration of the stimulus (*Kim et al., 2015*; *Nagel et al., 2015*). This implies that the circuitry connecting the *Or42a* and *Or42b* OSNs to PDM-DN must produce an inversion of the sign of the incoming olfactory stimulations to gate PDM-DN activity only when the OSN activity is low.

The two main upstream partners of PDM-DN are located in the lateral-horn (LH) region: LH-LN1 and LH-LN2 (*Figure 2B* and *Figure 2—figure supplement 1B–C*). These neurons form a feedforward motif where LH-LN1 outputs on LH-LN2 and PDM-DN whereas as LH-LN2 output on PDM-DN (*Figure 2C*, inset). Feedforward motifs (or feedforward loops) fulfill important regulatory functions in biological networks (*Alon, 2006*). Depending on the signs of the interactions between the LH-LN1, LH-LN2 and PDM-DN, this motif could act as pulse generator or a filter dampening off-responses of a sensory unit (*Shoval and Alon, 2010*). Given that the inputs of the *Or42a* and *Or42b* uPNs into this circuit will be correlated with changes in stimulus intensity, we conclude that either the synapses between LH-LN1 and PDM-DN or those between LH-LN2 and PDM-DN must be inhibitory. The absence of driver lines specific to the LH-LN1 and LH-LN2 neurons prevented us from resolving the sign of each interaction. In light of the ability of PDM-DN to deterministically trigger stops, we speculate that the 3-element feedforward circuit in the LH must represent the neural correlate of the action selection underpinning the sensorimotor control of the onset of reorientation maneuvers (*Figure 7B–C*). Future work will be necessary to clarify how dynamic trains of sensory inputs are converted into the transient activity of PDM-DN. In unpublished experiments, we attempted to

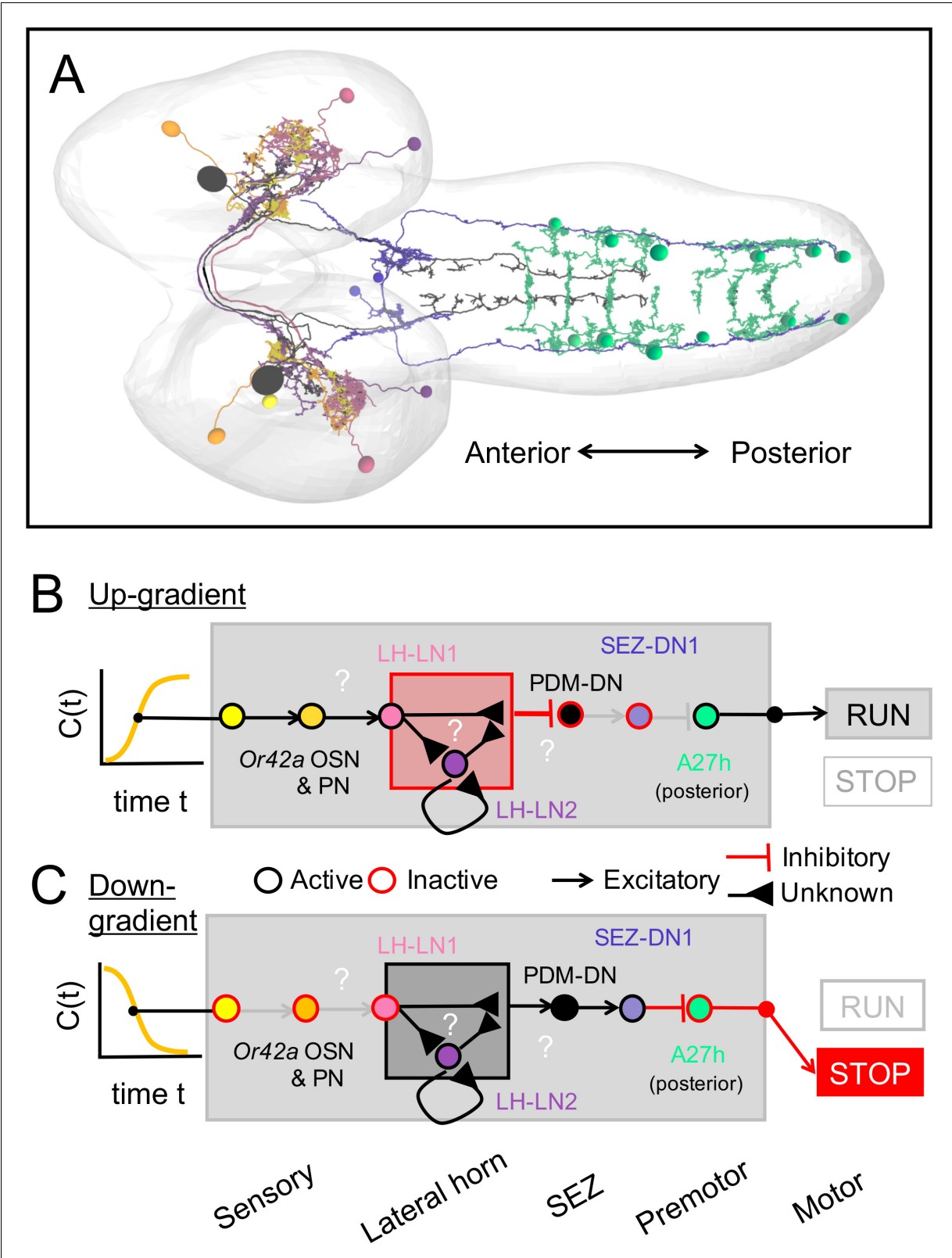

**Figure 7.** Sensorimotor circuit triggering the release of reorientation maneuvers during larval chemotaxis. (**A**) Key partners of the sensorimotor pathway bridging the *Or42a*-expressing olfactory sensory neuron (OSN, yellow) to the descending neuron PDM-DN (black), down to the A27h premotor circuit (green) in the ventral nerve cord. (**B**) Putative sensorimotor transformation of positive changes in odor concentration (C(t)) during up-gradient runs. Positive gradients detected by the *Or42a* OSN promote the OSN activity, which is expected to strongly active its cognate uniglomerular projection

*Figure 7 continued on next page*

*Figure 7 continued*

neuron (uPN, orange). Due to the cholinergic nature of olfactory projection neurons, excitation of the *Or42a* uPN promotes the activity of the LH-LN1 neuron (pink) located in the lateral horn region. In turn, the activity of LH-LN1 controls both the LH-LN2 and PDM-DN neuron. We speculate that the 3-element feedforward motif formed by LH-LN1, LH-LN2 and PDM-DN converts the activity of *Or42a* uPN into an inhibition of PDM-DN. As PDM-DN is cholinergic, the lack of activity of PDM-DN is expected to leave its downstream GABAergic partner SEZ-DN1 (blue) inactive. As a result, the sensorimotor pathway mediated by PDM-DN has no inhibitory effect on the A27h premotor circuit (green), which promotes forward peristalsis (runs). As indicated in the legend, neurons with a contour in black are thought to be excited. Neurons with a contour in red are thought to be inhibited or inactive. (**C**) Same as B upon detection of negative odor changes during down-gradient runs. Negative odor gradients inhibit the activity of *Or42a* OSN, which is thought to leave the *Or42a* uPN inactive. As a result, we speculate that the 3-element feedforward motif formed by LH-LN1, LH-LN2 and PDM-DN converts the inactivity of *Or42a* uPN into an excitation of PDM-DN. In turn, the firing activity of the cholinergic PDM-DN is thought to excite SEZ-DN1. Due to the GABAergic nature of SEZ-DN1, excitation of PDM-DN and SEZ-DN1 represses the activity of the A27h premotor circuit (green) in the posterior segments of the ventral nerve cord (VNC), thereby triggering an interruption in forward peristalsis (stop). Videos.
DOI: https://doi.org/10.7554/eLife.38740.024

characterize the response of PDM-DN to optogenetically-controlled activation of peripheral OSNs in brain explants. In spite of multiple attempts, these experiments were unsuccessful at producing reliable patterns of PDM-DN activity — a negative result that suggests that the absence of proprioceptive feedback in brain explants precludes the proper function of PDM-DN. Imaging the activity of PDM-DN in freely behaving animals might overcome this limitation in the future (*Karagyozov et al., 2017*).

Optogenetically-controlled activation of PDM-DN produces two distinct motor responses: (1) a cessation of forward peristalsis inducing a switch from running to stopping (*Figure 3E and F*) and (2) exploratory head movements followed by a turn (*Figure 3H and I*). The release of these two actions appears to be part of a hierarchy. The termination of peristalsis is near immediate (*Figure 3E* and *Figure 3—figure supplements 4A* and *5A*) and largely stereotypical across trials (*Figure 3—figure supplement 4C*). By contrast, the release of head casting takes a couple of seconds (*Figure 3H* and *Figure 3—figure supplement 4B*). Significant inter-trial variability is observed for the head-casting behavior (*Figure 3—figure supplement 4D*). Part of the variability in the head-casting behavior is reflected in the idiosyncratic nature of asymmetrical contractions in the thoracic and anterior abdominal segments, which might be influenced by experience-dependent factors (*Figure 3—figure supplement 5*). In agreement with a recent study in adult flies (*Cande et al., 2017*), our results argue that a single descending neuron can contribute to the sensorimotor control of different actions. While stopping behavior limits overshoots of the odor source, sensory-dependent release of patterns of head casts enable the larva to scan the local odor gradients to reorient toward the direction of higher concentrations (*Figure 3D*, right panel and *Gomez-Marin et al., 2011*). By taking advantage of the EM reconstruction, we built a circuit diagram of the main partners downstream from PDM-DN and sought to delineate the neural pathway actuating stops in forward locomotion and head-casting behaviors (*Figure 7*).

Forward locomotion through peristalsis rely on the coordinated inter-segmental propagation of waves of muscle contractions from the posterior (tail) to the anterior end (head) of the body segments (*Heckscher et al., 2012*). This cyclic behavior emerges from the activity of a network of premotor neurons that spans the entire set of abdominal segments of the VNC. The cessation of forward peristalsis (stopping behavior) can be accomplished in at least three different ways: (i) by preventing the initiation of forward peristaltic waves; (ii) by inhibiting forward wave propagation or (iii) by the combination of both mechanisms. The following observations support a model in which PDM-DN mediates stopping behavior by inhibiting the initiation of forward peristaltic waves in the posterior abdominal segments of the VNC (mechanism i). First, the analysis of peristaltic wave propagation in freely behaving larvae responding to PDM-DN activation suggested that optogenetic activation of PDM-DN is insufficient to inhibit wave propagation once the wave has already been initiated (*Figure 4A–C*). Second, a detailed tracking of the segmental contractions in restrained larvae showed that PDM-DN activation can inhibit wave initiation in the posterior segments, but not in the anterior segments (*Figure 4E–F*). The 4th abdominal segment (A4) appears to be the 'hinge' region beyond which PDM-DN fails to inhibit the wave propagation. Third, similar observations were made by using calcium imaging to monitor fictive patterns of locomotion in isolated CNS preparations in response to PDM-DN activation (*Figure 4G–H*). In agreement with published results related to the sequential ablation of abdominal segments in CNS explants (*Pulver et al., 2015*), we found that

PDM-DN activation blocks forward wave propagation most effectively between the 5th and the 7th abdominal segments (A5-A7). We concluded that PDM-DN might specifically target the pre-motor circuit responsible for the initiation of forward locomotion in the most posterior segments, while enabling asymmetrical contractions of the thoracic and anterior abdominal segments to scan the local odor gradient through head casts and to implement turning maneuvers (*Lahiri et al., 2011*; *Berni, 2015*).

What are the neural mechanisms underlying the inhibitory action of PDM-DN on the pre-motor system of the larva? The EM reconstruction of the downstream partners of PDM-DN revealed that that PDM-DN synapses onto a set of local and descending interneurons in the SEZ region (*Figure 5*). We have previously shown that the SEZ comprises a subset of neurons that are necessary and sufficient to trigger reorientation maneuvers in response to multi-sensory stimuli (*Tastekin et al., 2015*). In agreement with this finding, the activity of the SEZ region correlates with the initiation and execution of forward peristaltic waves (*Lemon et al., 2015*). The SEZ of the larva also participates in the control of switches between feeding and crawling behaviors (*Schoofs et al., 2014*). More generally, the SEZ acts as a pre-motor hub that integrates dynamic sensory inputs and coordinate the release of specific motor programs (*Libersat and Gal, 2013*; *Hampel et al., 2015*). We found that the PDM-DN relays its 'command' through a small set of larval descending neurons that have their dendritic harbors in the SEZ. The main downstream partner of PDM-DN is a descending neuron SEZ-DN1 also known as Pair-1 (*Figure 5—figure supplement 1C* and *Figure 7A*), which projects to the posterior abdominal segments (*Figure 5* and *Figure 5—figure supplement 1C*). SEZ-DN1 synapses on a circuit of segmentally repeated excitatory premotor neurons (A27h) that is involved in the propagation of forward peristaltic waves (*Fushiki et al., 2016*). The synapses between SEZ-DN1 and the A27h circuit are mainly restricted to the posterior abdominal segments A5-A7. We propose that PDM-DN inhibits the initiation of forward peristaltic waves via SEZ-DN1. Neurotransmitter profiling of PDM-DN demonstrated that this neuron is excitatory. Given the inhibitory effect of PDM-DN activation on peristalsis, we hypothesized that SEZ-DN1 must inhibit the activation of the A27h neurons. In agreement with a companion study (*Carreira-Rosario et al., 2018*), co-labeling of SEZ-DN1 with GABA antibody corroborated the inhibitory nature of this neuron (*Figure 5C*). Like PDM-DN, optogenetic activation of SEZ-DN1 is sufficient to evoke stopping patterns of forward locomotion (*Figures 5D* and *6A–B*). In imaging experiments where fictive motor waves were visualized with GCaMP6f, we demonstrated that acute optogenetic activation of SEZ-DN1 interrupted the propagation of peristaltic waves (*Figure 6C–D*). The connectivity between PDM-DN and SEZ-DN1 was further established by eliciting robust patterns of SEZ-DN1 activity upon optogenetic activation of PDM-DN (*Figure 6E–F*). Although we cannot exclude that parallel pathways downstream from PDM-DN contribute to the induction of stopping behavior, we propose that SEZ-DN1 is a descending neuron that can trigger stopping behavior by inhibiting forward wave initiation in the most posterior abdominal segments of the VNC — a conclusion supported by recent work (*Carreira-Rosario et al., 2018*). The bilateral projection of PDM-DN on the left and right SEZ-DN1 neurons (*Figure 5—figure supplement 1C*) explains the ability of unilateral optogenetic activation of PDM-DN to produce symmetrical block in peristalsis (*Figure 3—figure supplement 3B*).

Having identified SEZ-DN1 as the main actuator of pauses upon PDM-DN gain-of-function, we turned to the second phenotype triggered by PDM-DN activation: the release of head casting behavior in preparation of turning maneuvers. By reviewing the downstream partners of PDM-DN, we discovered that at least two pathways might be implicated in the release of head-casting behavior: SEZ-LN1 and SEZ-DN2 (*Figure 5A*). The SEZ-LN1 neuron lies upstream from an uncharacterized premotor neuron, which gives asymmetrical input into anterior RP neurons that control either dorsal or ventral muscles on the body walls (*Figure 5—figure supplement 2*). We speculate that asymmetrical contractions of dorsal and ventral muscles in the anterior segments facilitate head casting and turning behaviors. Likewise, SEZ-DN2 gives input into pre-motor neurons upstream of prothoracic accessory nerve (PaN) motor neurons that are thought to mediate head tilting (*Hückesfeld et al., 2015*) — a behavior frequently observed during reorientation maneuvers. Due to the absence of sparse driver lines that specifically label SEZ-LN1 and SEZ-DN2, we could not examine the function of these neurons.

In summary, the present study describes the reconstruction of a sensorimotor pathway from the peripheral sensory system down to the motor system (*Figure 7A*). We identified and characterized a descending neuron, PDM-DN, which plays a central role in controlling the release of reorientation

maneuvers based on the integration of sensory experience. This command-like neuron illustrates the versatility of the behavioral control descending neurons are capable of (*Cande et al., 2017*). While stopping behavior is deterministically triggered by PDM-DN activation, the release of head casting and turning behaviors was context-dependent. Our results argue that these two behavioral programs — stopping and head-casting — are partly controlled by independent pathways under the control of different descending neurons. EM reconstruction and functional analysis revealed that PDM-DN employs distinct SEZ and abdominal interneurons to differentially regulate stopping and head casting/turning behaviors. Considering the striking similarity between the navigation algorithms that control orientation to different sensory modalities (thermotaxis, phototaxis and chemotaxis) (*Louis et al., 2008*; *Luo et al., 2010*; *Kane et al., 2013*), it is plausible that PDM-DN contributes to the control of stopping behavior elicited by visual and thermal signals too. Alternatively, multiple parallel descending pathways might contribute to the sensory control of switches between running and stopping. To produce a coherent motor outcome, 'commands' arising from these different pathways would have to be integrated downstream from PDM-DN. Where and how this integration takes place remains a mystery that has now become experimentally tractable.

# Materials and methods

## Key resources table

| Reagent type | Designation | Source or reference | Identifiers | Additional information |
|---|---|---|---|---|
| Fly line | R23E07-p65.AD | Bloomington drosophila stock center | Rrid: BDSC_70131 | Landing site: attP40 |
| Fly line | VT002081-GAL4.DBD | Bloomington Drosophila Stock Center | RRID: BDSC_73398 | Landing site: attP2 |
| Fly line | UAS-TNTE | (*Tastekin and Louis, 2017*) | NA | Second chromosome |
| Fly line | UAS-TNTG | Bloomington Drosophila Stock Center | RRID: BDSC_28838 | Second chromosome |
| Fly line | 20XUAS-IVS-Cs Chrimson.mVenus | Bloomington Drosophila Stock Center | RRID: BDSC_55136 | Landing site: attP2 |
| Fly line | 20XUAS-IVS-Cs Chrimson.mVenus | Bloomington Drosophila Stock Center | RRID: BDSC_55134 | Landing site: attP18 |
| Fly line | R57C10-FLP2::PEST | (*Nern et al., 2015*) | NA | Gift from Aljoscha Nern |
| Fly line | R57C10-FLPL attp8; 10xUAS(FRT.stop)myr ::smGdP-HA VK00005; 10xUAS(Frt.stop)myr ::smGdP-V5-THS-10XUAS(FRT.stop)myr ::smGdP-FLAG su(Hw)attP1 | Bloomington Drosophila Stock Center | RRID: BDSC_64087 | Gift from Aljoscha Nern |
| Fly line | CG9887-LexA su(Hw)attP8 | NA | NA | Julie Simpson |
| Fly line | 20XUAS(Frt.stop) Chrimson::mVenus attP2 | NA | NA | Gift from Vivek Jayaraman |
| Fly line | 20XUAS-IVS-CsChrimson.mVenus attP18; CG9887-LexA su(Hw)attP8; 12XlexAop2-IVS-GCaMP6f-p10 su(Hw)attP5 | NA | NA | Gift from Stefan Pulver |

*Continued on next page*

*Continued*

| Reagent type | Designation | Source or reference | Identifiers | Additional information |
|---|---|---|---|---|
| Fly line | R75C02-GAL4 | (*Li et al., 2014*) available at Bloomington Drosophila Stock Center | RRID: BDSC_39886 | Landing site: attP2 |
| Fly line | R75C02-LexA/Cyo | (*Li et al., 2014*) available at Bloomington Drosophila Stock Center | RRID: BDSC_54365 | Landing site attP40 |
| Fly line | ChAT RNAi | Vienna Drosophila Resource Center | RRID: FlyBase_FBst0463844 | |
| Fly line | 13xLexAop2-IVS-GCaMP6f-p10 50.693 in VK00005, 20xUAS-CsChrimson-mCherry-trafficked in su(Hw)attp1 | NA | NA | Gift from Vivek Jayaraman |
| Odor | Ethyl butyrate | Millipore Sigma, MO, USA | Cat# 19230 | CAS#: 105-54-4 |
| Odor | Ethyl acetate | Millipore Sigma, MO, USA | Cat# 319902 | CAS#: 141-78-6 |
| Chemical | Paraffin oil | Millipore Sigma, MO, USA | Cat# 18512 | CAS#: 8012-95-1 |
| Chemical | PBS | Millipore Sigma, MO, USA | Cat# P4417 | |
| Chemical | Agarose | Lonza, Basel, Switzerland | Cat# 50001 | |
| Chemical | Bacto Agar | Thermo Fisher Scientific, MA, USA | Cat# DF0812-07-1 | |
| Chemical | Vectashield | Vector Laboratories, CA, USA | Cat# H1000 | |
| Chemical | Poly-L-lysine solution | Millipore Sigma, MO, USA | Cat# P8920 | CAS#: 25988-63-0 |
| Chemical | Formaldehyde | Thermo Fisher Scientific, MA, USA | Cat# NC9658705 | |
| Chemical | All trans-retinal | Millipore Sigma, MO, USA | Cat# R2500 | CAS#: 116-31-4 |
| Chemical | Sucrose | Millipore Sigma, MO, USA | Cat# S0389 | CAS#: 57-50-1 |
| Chemical | Sodium Chloride | Millipore Sigma, MO, USA | Cat# S7653 | CAS#: 7647-14-5 |
| Chemical | Calcium Chloride | Millipore Sigma, MO, USA | Cat# 21115 | CAS#: 10043-52-4 |
| Chemical | Magnesium Chloride Hexahydrate | Millipore Sigma, MO, USA | Cat# P2670 | CAS#: 7791-18-6 |
| Chemical | Potassium Chloride | Millipore Sigma, MO, USA | Cat# P9571 | CAS#: 7447-40-7 |
| Chemical | TES | Millipore Sigma, MO, USA | Cat# T1375 | CAS#: 7365-44-8 |
| Chemical | Sylgard 184 | Dow Corning, MI, USA | Cat# 4019862 | |

*Continued on next page*

*Continued*

| Reagent type | Designation | Source or reference | Identifiers | Additional information |
|---|---|---|---|---|
| Antibody | Mouse anti-brp (nc82) | Development Studies Hybridoma Bank (DSHB), IA, USA | RRID: AB_2314866 | 1:25 dilution |
| Antibody | Mouse anti-synorf1 (3C11) | Development Studies Hybridoma Bank (DSHB), IA, USA | RRID: AB_2313867 | 1:25 dilution |
| Antibody | Rabbit anti-GFP | Thermo Fisher Scientific, MA, USA | RRID: AB_221569 | 1:500 dilution |
| Antibody | Mouse anti-GFP | Thermo Fisher Scientific, MA, USA | RRID: AB_221568 | 1:500 dilution |
| Antibody | Mouse anti-ChAT (ChAT4B1) | Development Studies Hybridoma Bank (DSHB), IA, USA | RRID: AB_528122 | 1:100 dilution |
| Antibody | Rabbit anti-Glutamate | - | RRID: AB_2490070 | 1:1000 dilution Gift from Hermann Aberle |
| Antibody | Rabbit anti-GABA | MilliporeSigma, MO, USA | RRID: AB_472652 | 1:2000 dilution |
| Antibody | Rabbit anti-HA-Tag (C29F4) | Cell Signaling Technology, MA, USA | RRID: AB_1549585 | 1:500 dilution |
| Antibody | Goat anti-Rabbit Alexa Fluor 488 | Thermo Fisher Scientific, MA, USA | RRID: AB_143165 | 1:500 dilution |
| Antibody | Goat anti-Rabbit DyLight 549 | Vector Laboratories, CA, USA | RRID: AB_2336407 | 1:500 dilution |
| Antibody | Goat anti-Mouse Texas Red | Jackson Immuno Research Laboratories, PA, USA | Cat# 115-075-003 | Discontinued 1:500 dilution |
| Antibody | Goat anti-Mouse FITC | Millipore Sigma, MO, USA | RRID: AB_259804 | 1:500 dilution |
| Serum | Goat Serum | Millipore Sigma, MO, USA | Cat# G9023 | 3% dilution |
| Consumable | 96 well plate lids with condensation rings | Millipore Sigma, MO, USA | Cat# L3661 | |
| Consumable | 96 well plate lids without condensation rings | Millipore Sigma, MO, USA | Cat# L3536 | |
| Consumable | 245 mm x 245 mm Petri dish | Millipore Sigma, MO, USA | Cat# CLS431111 | |
| Software | MATLAB 2015b | Mathworks, MA, USA | RRID: SCR_001622 | |
| Software | LabView | National Instruments, TX, USA | RRID: SCR_0014325 | |
| Software | Fiji/ImageJ | Fiji | RRID: SCR_002285 | |

## Fly strains

Fly strains used in this study are listed in Key resources table. Flies were raised on standard cornmeal medium at 22–23°C on 12 hr-day: 12-hr-night cycle. PDM-DN is targeted by using the Split-GAL4 combination: y⁻w⁻; R23E07-p65.AD; VT002081-GAL4.DBD (PDM-DN-GAL4). SEZ-DN1 is targeted by using R75C02-GAL4. Wild-type background for Split-GAL4 lines is y⁻w⁻ flies with empty attP40 and attP2 landing sites (y⁻w⁻; attP40, attP2). Wild-type background for GAL4 lines is w⁻ flies with empty attP2 landing site. For control experiments, we used w⁻ flies as wild-type background crossed to the UAS-TNTE effector line.

## Histology

Larval CNS were dissected and fixed in 4% formaldehyde (NC9658705, Thermo Fisher Scientific, MA, USA) in PBS (P4417, MilliporeSigma, MO, USA) for 1 hr at room temperature. After 3 times 15 min washes in PBS + 1% Triton X-100 (9284, TX, MilliporeSigma, MO, USA), the tissues were blocked for 1 hr at room temperature with 3% normal goat serum (G9023, MilliporeSigma, MO, USA) in 1% PBS-TX. The tissues were incubated with primary antibodies overnight at 4 °C. After 4 times 20 min washes in 1% PBS-TX, the samples were incubated with secondary antibodies for 4 hr at room temperature. Tissues were washed 4 times for 20 min with 1% PBS-TX followed by a short wash with PBS. The CNS were mounted in Vectashield mounting medium (H1000, Vector Laboratories, CA, USA) on lysine-coated (P8920, MilliporeSigma, MO, USA) coverslips and kept at 4 °C until imaged. Imaging was performed by using either a Leica SP5 or a Leica SP8 confocal microscope (Leica Microsystems, Wetzlar, Germany). Image rendering was performed with Fiji (ImageJ, http://fiji.sc/). Background correction, brightness and contrast adjustments, 3D rendering and maximum projections were performed by using Fiji. Antibodies and dilutions used in this study are listed in Key resources table. In order to detect CsChrimson::mVenus, we used conventional GFP antibodies (RRID: AB_221568 or RRID: AB_221569, Thermo Fisher Scientific, MA, USA). For MCFO experiments, we used an antibody against HA-tag (RRID: AB_1549585, Cell Signaling Technology, MA, USA). To label the neuropil, we used a combination of anti-brp (RRID: AB_2314866, Development Studies Hybridoma Bank, IA, USA) and anti-synorf1 (RRID: AB_2313867, Development Studies Hybridoma Bank, IA, USA) antibodies.

## Loss-of-function screen using Split-GAL4 driver lines

Split-GAL4 lines were crossed to UAS-TNTG (RRID: BDSC_28838, Bloomington Drosophila Research Center). Prior to tracking, 3rd instar larvae were kept in 15% sucrose (S0389, MilliporeSigma, MO, USA) for 20 min. After a quick wash in distilled water, twenty larvae were gently placed on 4% agar (DF0812-07-1, Thermo Fisher Scientific, MA, USA) slabs in 245 mm x 245 mm square Petri dishes using a small paint brush. Four odor droplets of 8 μl of 15mM ethyl butyrate (19230, MilliporeSigma, MO, USA) diluted in paraffin oil (Sigma-Aldrich) were pipetted inside the lid of the Petri dish (Figure 1—Figure supplement 1). Shortly after the lid was closed on the Petri dish, larvae were tracked for 6 min using Multi Worm Tracker software (http://sourceforge.net/projects/mwt). Behavioral data were analyzed using custom-written MATLAB scripts (https://github.com/LabLouis/eLife2018_PDM-DN; copy archived at https://github.com/elifesciences-publications/eLife2018_PDM-DN). The odor gradient was numerically simulated by using the analytical solution of an idealized diffusion problem from four point-odor sources in an environment with absorbing boundary conditions. At any given position in the assay $(x, y, z)$, the odor concentration $C(r, t)$ was modeled as $\frac{\gamma}{4 \pi D r} \cdot \mathrm{erfc}(\frac{r}{\sqrt{4 D t}})$ where $\mathrm{erfc}(x)$ is the error function defined as $\int_0^x e^{-s^2} ds$ and $r$ denotes the distance to the source with coordinate $(x_s, y_s, z_s)$: $r = \sqrt{(x - x_s)^2 + (y - y_s)^2 + (z - z_s)^2}$. Since we were only interested in reconstructing the geometry of the gradient and not its absolute concentration, the value of the flux of odor from the source was set arbitrarily equal to 1. The diffusion constant of the odor was approximated to 0.025 cm²/min.

## Single larva tracking experiments

In these experiments, UAS-TNTE was used to silence neural activity (*Tastekin et al., 2015*). Single larva tracking was performed as it was described before (*Gomez-Marin et al., 2011*). Briefly, 3rd instar larvae were kept in 15% sucrose for 20 min prior to tracking. Outer surface of 96 well plate

lids without condensation rings were covered with 25 ml of 3% agarose (Seakem-LE, Lonza, Basel, Switzerland). 5 µl of odor (100 µM ETB or 1 mM EtA) was placed in one of the condensation rings in the middle of the 96 well plate lid with condensation rings (same ring location was used for all trials). The lid with the odor droplet was inverted on the agarose slab. The distance between the top of the lid and the agarose layer was sufficient to prevent any contact between the larva and the odor source (*Gomez-Marin and Louis, 2012*). A single larva was placed on the agarose surface approximately three wells away from the odor source. The larva was gently oriented towards the odor source and subsequently tracked for 5 min. The odor gradient was quantified by using a FT-IR spectrometer (Tensor 27, Bruker, MA, USA) as described elsewhere (*Tastekin et al., 2015*). Behavioral data were acquired, analyzed and quantified by using custom written MATLAB scripts (*Tastekin et al., 2015*) (updated version available via https://github.com/LabLouis/eLife2018_PDM-DN; copy archived at https://github.com/elifesciences-publications/eLife2018_PDM-DN).

## Gain-of-function manipulations for freely behaving larvae

GAL4/Split-GAL4 driver lines were crossed to 20XUAS-IVS-CsChrimson::mVenus in attP2 landing site (Bloomington Drosophila Research Center, RRID: BDSC_55136). For the gain-of-function experiments with ChAT-RNAi, 20XUAS-IVS-CsChrimson::mVenus in attP18 landing site was used (Bloomington Drosophila Research Center, RRID: BDSC_55134). Flies were kept in complete darkness on food supplemented with 0.5 mM all trans-retinal (R2500, MilliporeSigma, MO, USA). For optogenetic stimulation experiments, we used a red LED with peak emission at 625 nm (PLS 0625–030 s, Mightex Systems, Toronto, Canada). Single larvae were placed on a 2.5% agarose slab and tracked for 5 min using a closed-loop tracker (*Schulze et al., 2015*). For most of the experiments, we applied eight light flashes that were 6 s long. Consecutive flashes were separated by 30 s of no stimulation. For longer activation of the PDM-DN neuron, we applied 6 flashes of 20 s interspersed by 30 s of no stimulation. For head cast/turn-triggered activation experiments (*Figure 3—figure supplement 2G–H*), head casts/turns were classified online as described before (*Schulze et al., 2015*) and light flashes were kept on as long as the larva remained in head casting/turning mode.

## Stochastic gain-of-function assay

To unilaterally express CsChrimson::mVenus in the PDM-DN neuron, R57C10-Flp2::PEST;+; 20XUAS (FRT.stop)Chrimson::mVenus attP2 was crossed to (PDM-DN-GAL4). Flp2::PEST is a low activity flippase that allowed us to produce unilateral patterns of stochastic expression of CsChrimson::mVenus on one side of the brain (*Nern et al., 2015*). The larvae were blindly screened for gain-of-function behavior in the closed-loop tracker before their nervous system was isolated and immunostained against CsChrimson::mVenus.

## Gain-of-function manipulations for restricted larvae

Larvae at the 3$^{rd}$ instar developmental stage were pinned down on a PDMS slab (Sylgard 184 Silicone Elastomer Kit, Dow Corning) using stainless steel pins (26002–10, Fine Science tools, Vancouver, Canada). The pins were placed at the anterior end of the animal between the mouth hooks and at the posterior end of the animal between the tail spiracles. The segmental contractions were tracked by visualizing the denticle bands on the ventral side of the larva. Videos were recorded at a resolution of 1280 × 720 pixels and a rate of 20 Hz by using a USB3.0 CMOS camera (Grasshopper 3–41 C6M-C, Point Grey, Richmond, Canada). The movements of the denticle bands were analyzed by using Fiji and custom-written scripts in MATLAB. For optogenetic stimulations, we used a red LED with peak emission at 625 nm (PLS 0625–030 s, Mightex Systems, Toronto, Canada). We applied 6 s long flashes interspaced by 20 s. The camera and the LED were triggered with a NI-DAQ (NI USB-6525, National Instruments, TX, USA) and a custom written LabVIEW software (LabVIEW, National Instruments, TX, USA).

## Analysis of motor neuron activity and fictive locomotion

GAL4/Split-GAL4 lines were crossed to 20XUAS-IVS-CsChrimson.mVenus attP18; CG9887-LexA su (Hw)attP8; 12XlexAop2-IVS-GCaMP6f-p10 su(Hw)attP5 (gift from Stefan Pulver). CG9887-LexA (gift from Julie Simpson) targets GCaMP6f expression to glutamatergic neurons including all motor neurons. The central nervous system (CNS) of larvae at the 3$^{rd}$ instar developmental state were

dissected in cold saline composed of 135 mM NaCl (S7653, MilliporeSigma, MO, USA), 5 mM KCl (P9571, MilliporeSigma, MO, USA), 2 mM CaCl2 (21115, MilliporeSigma, MO, USA), 4 mM $MgCl_2$ (P2670, MilliporeSigma, MO, USA), 5 mM TES (T1375, MilliporeSigma, MO, USA) and 36 mM sucrose (S0389, MilliporeSigma, MO, USA). Isolated CNS were mounted on lysine-coated coverslips and covered with the saline solution. Imaging of motor neuron activity was performed with a Leica SP5 upright microscope using a 25X water immersion objective (NA. 95, 15506374, Leica Microsystems, Wetzlar, Germany) or 20X multi-immersion objective with a correction collar (NA 0.75, 15506343, Leica Microsystems, Wetzlar, Germany). Optogenetic activation of PDM-DN or SEZ-DN1 was performed by using a red LED with peak emission at 625 nm (PLS 0625–030 s, Mightex Systems, Toronto, Canada). The microscope and the LED were triggered with NIDAQs (NI USB-6229 and NI USB-6009, National Instruments, TX, USA) using custom written MATLAB scripts. Very low powers of Argon laser (488 nm) were used to excite GCaMP6f while avoiding the activation of CsChrimson at this wavelength. In the absence of 625 nm LED stimulus, waves of motor-neuron activity were observed (*Figures 5* and *6*) suggesting that the Argon laser did not interfere with CsChrimson activation. Recordings were performed at 4–5 Hz. The imaging data were analyzed using Fiji and custom written MATLAB scripts (https://github.com/LabLouis/eLife2018_PDM-DN; copy archived at https://github.com/elifesciences-publications/eLife2018_PDM-DN). Changes in the fluorescence signal were quantified as follows: region of interests (ROIs) were chosen for each segment on both sides of the VNC (see illustration in *Figure 4G*). The mean pixel intensity was calculated for each ROI. Baseline fluorescence intensity $F_o$ was calculated as mean pixel intensity of the given ROI over the whole time series. For each time frame $F_i$, signals are computed as the relative change in fluorescence intensity from the baseline: $\Delta F/F = (F_i-F_0)/F_0$ (*Jia et al., 2011*). The $\Delta F/F$ signal was filtered using Savitzky-Golay filtering with third order polynomial (frame length: 15).

## Imaging SEZ-DN1 activity in response to PDM-DN activation

Imaging experiments were performed with the following cross: 75C02-LexA/Cyo-Dfd::YFP; 13xLex-Aop2-IVS-GCaMP6f-p10 50.693 in VK00005, 20xUAS-CsChrimson-mCherry-trafficked in su(Hw)attp1 was crossed to PDM-DN-GAL4. The CNS of larvae at the 3$^{rd}$ instar developmental stage were dissected in cold saline (identical to the saline used in motor neuron imaging) and placed dorsal side down on a lysine-coated coverslip. We used an Ultima II two-photon scanning microscope (Bruker, MA, USA) and a Vision II laser (Coherent, CA, USA). For imaging we used two-photon excitation wavelength of 920 nm with the power at the sample never exceeding 10 mW. CsChrimson was activated with a LED (M590L3-C1, Thorlabs, NJ, USA) with peak wavelength at 590 nm and a 605/55 bandpass filter. To feed the CsChrimson excitation light into the light path, we used a custom dichroic dichroic (zt488-568, Chroma, VT, USA). The power of the LED from the objective was ~30 $\mu W/mm^2$. Optogenetic activation stimuli were delivered in 30 Hz pulse trains (2 ms on, 31 ms off) lasting for 1 s. For trial the neuron was observed for approximately 20 s. Each trial was repeated 5 times. Functional imaging was analyzed by using custom written Python scripts (https://github.com/LabLouis/eLife2018_PDM-DN; copy archived at https://github.com/elifesciences-publications/eLife2018_PDM-DN). A ROI was chosen to contain only the axonal branching of the neuron of interest (*Figure 6E*). The signal was calculated as described previously (*Jia et al., 2011*). No noise filtering was applied. For each time frame $F_i$, signals were computed as the relative change in fluorescence intensity from the baseline: $\Delta F/F = (F_i-F_0)/F_0$. Baseline fluorescence $F_0$ was defined as the mean pixel intensity of a 2 s time window preceding the optogenetic stimulus.

## EM reconstruction using CATMAID

EM reconstruction of neurons and annotations of synapses were performed as described previously (*Ohyama et al., 2015*) using web-based software CATMAID (*Saalfeld et al., 2009*). An iterative reconstruction method was applied to reconstruct and annotate pre- and post-synaptic partners (*Ohyama et al., 2015*; *Schneider-Mizell et al., 2016*).

## Statistical analysis

Normality was tested using Lilliefors test. If distributed normally, data were analyzed with ANOVA test. When the Lilliefors test rejected normality, we used Wilcoxon ranksum test followed by Bonferroni correction. For paired data in *Figure 3—figure supplement 3G*, *Figure 4H* and *Figure 6D*, the

Wilcoxon signed-rank test was used. For binomially distributed data, Clopper Pearson method was used to determine 95% confidence intervals. The cross correlation analysis was carried with the built-in *corrcoef* function of MATLAB_R2015b. To compare proportions in *Figure 4F*, we used a standard Z-test. More information about the statistical procedures can be found in the Transparency Report. The list of p-values corresponding to each statistical test of the study can be found in *Supplementary file 1*.

## Acknowledgements

We thank the Louis and the Simpson lab for discussions throughout the project. We are grateful to CDoe, D Goldschmidt and J Kenchel for comments on the manuscript. This work was initiated as part of the multi-lab Larval Olympiad conducted at the Janelia Research Campus. We are in debt to the work of S Reid (Louis lab) during the initial phase of the screen. We thank I Andrade, A Fushiki, J Jonaitis, I Larderet, P Schegel, C Schneider-Mizell and M Zwart for contributing to the EM reconstruction. We thank H Aberle for glutamate antibodies, as well as V Jayaraman, A Nern, S Pulver, G Rubin and J. Simpson for sharing fly lines. We thank V Jayaraman and R Francoville for training and access to the functional imaging setup. ML and DT acknowledges support of the Spanish Ministry of Economy and Competitiveness (MICINN and BFU2011-26208), 'Centro de Excelencia Severo Ochoa 2013–2017', the CERCA Programme/Generalitat de Catalunya, the EMBL/CRG Systems Biology Program and the University of California, Santa Barbara. IT was supported by the Marie Curie FP7 Programme through FLiACT (ITN). AK was supported by the 'La Caixa' International PhD Programme. JT, MZ and AC acknowledge funding from the Howard Hughes Medical Institute.

## Additional information

### Funding

| Funder | Grant reference number | Author |
| --- | --- | --- |
| Spanish Ministry of Economy and Competitiveness | BFU2011-26208 | Ibrahim Tastekin<br>Avinash Khandelwal<br>David Tadres<br>Nico D Fessner<br>Matthieu Louis |
| European Commission | Marie Curie FP7 Programme ITN-FLiACT | Ibrahim Tastekin<br>Matthieu Louis |
| Spanish Ministry of Economy and Competitiveness | BFU2014-55059 | Ibrahim Tastekin<br>Avinash Khandelwal<br>David Tadres<br>Nico D Fessner<br>Matthieu Louis |
| "la Caixa" Foundation | International PhD program | Avinash Khandelwal |
| University of California, Santa Barbara | | David Tadres<br>Matthieu Louis |
| Howard Hughes Medical Institute | | James W Truman<br>Marta Zlatic<br>Albert Cardona |

The funders had no role in study design, data collection and interpretation, or the decision to submit the work for publication.

### Author contributions

Ibrahim Tastekin, Conceptualization, Resources, Data curation, Software, Formal analysis, Investigation, Visualization, Methodology, Writing—original draft; Avinash Khandelwal, Data curation, Formal analysis, Investigation; David Tadres, Conceptualization, Software, Formal analysis, Investigation, Visualization, Methodology, Writing—original draft; Nico D Fessner, Data curation, Investigation; James W Truman, Resources, Methodology; Marta Zlatic, Resources, Funding acquisition, Project administration; Albert Cardona, Data curation, Supervision, Funding acquisition, Investigation,

Methodology; Matthieu Louis, Conceptualization, Formal analysis, Supervision, Funding acquisition, Visualization, Methodology, Writing—original draft, Project administration

## Author ORCIDs

Ibrahim Tastekin http://orcid.org/0000-0003-3661-9115
David Tadres http://orcid.org/0000-0002-7570-0162
James W Truman http://orcid.org/0000-0002-9209-5435
Albert Cardona http://orcid.org/0000-0003-4941-6536
Matthieu Louis http://orcid.org/0000-0002-2267-0262

## Decision letter and Author response

Decision letter https://doi.org/10.7554/eLife.38740.028
Author response https://doi.org/10.7554/eLife.38740.029

## Additional files

### Supplementary files

• Supplementary file 1. Tables summarizing the conditions and results of the statistical tests conducted throughout the manuscript.
DOI: https://doi.org/10.7554/eLife.38740.025

• Transparent reporting form
DOI: https://doi.org/10.7554/eLife.38740.026

### Data availability

Scripts for data analysis, source data files for the behavioral and imaging experiments have been made available on the GitHub account of the Louis lab (https://github.com/LabLouis/eLife2018_PDM-DN; copy archived at https://github.com/elifesciences-publications/eLife2018_PDM-DN).

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
