## [Decision Letter]

Thank you for submitting your article "Sensorimotor pathway controlling stopping behavior during chemotaxis in the *Drosophila melanogaster* larva" for consideration by *eLife*. Your article has been reviewed by three peer reviewers, including Ronald L Calabrese as the Reviewing Editor and Reviewer #1, and the evaluation has been overseen by K VijayRaghavan as the Senior Editor. The following individual involved in review of your submission has agreed to reveal their identity: Jimena Berni (Reviewer #3).

The reviewers have discussed the reviews with one another and the Reviewing Editor has drafted this decision to help you prepare a revised submission.

Summary:

In this manuscript, the authors present a thorough analysis of circuit elements that mediate in olfactory guided stops and head casts associated with chemotaxis to an attractive odor in *Drosophila* larvae. In particular they concentrate on the role of a newly identified olfactory descending neuron, PDM-DN (bilateral pair), originating in the Lateral Horn region of the Brain Lobes and descending to the Ventral Nerve Cord. They use previous em reconstructions of the larval nervous system to identify upstream circuitry leading from olfactory receptor neurons, through specific olfactory Projection Neurons, through specific LH neurons to the PDM-DNs. They use the same reconstructions to further trace downstream circuitry focusing mainly on a path from the PDM-DNs onto the SubEsophageal Zone DN1 which then projects onto the previously characterized segmentally repeated (abdominal segments of the VNC) excitatory premotor interneurons A27h to which they connect in the posterior segments. The manuscript makes full use of current behavioral, intersectional genetics and optogenetic techniques available for circuit analyses of behavior in *Drosophila* and is guided by an em reconstruction of upstream and downstream synaptic targets. They perform critical necessity and sufficiency tests for the PDM-DNs and some downstream elements. The level of circuit analyses is impressive and it clearly shows that the PDN DNs play a critical role in the halts and head casts that accompany down-gradient runs in attractive chemotaxis. They also present a very convincing analysis of how PDM DN (and SEZ DN1) activation halts locomotory waves of contraction and fictive locomotory activity recorded with Ca indicator in the posterior abdominal segments of the VNC. The work is truly elegant and pushes the technology available. The elegance of both the analysis and the circuitry uncovered for mediating down-gradient run associated halts and head casts will be of wide interest.

The writing is clear and the figures are well organized, and clear.

Essential revisions:

1) Fundamental for the interpretation of all the results presented in the paper is the demonstration that the PDM-DNs plays a role in chemotaxis and not in crawling itself. The authors should show that when there is no odor and PDM-DNs are blocked, the larvae behave as their parental genetic controls. The experiment in Figure 3—figure supplement 1 no odor, should be performed with the same number of animals (30) required to see a significant difference in the chemotaxis experiment. If the behavior is not as the controls, then the entire paper will have to be reinterpreted and rewritten.

2) The sufficiency and necessity tests performed and the associated analyses are very impressive, and the data are convincing; however, they do not show necessity for halts or for head casts. Rather, they show that the PDN DNs play a critical role in halts and head casts associated with down-gradient runs but are not essential for such halts and head casts. Similarly, the sufficiency tests do not present an absolute result indicating that activation of the neuron produces a halt. It is not important to claim necessity or sufficiency but rather to present the impressive circuit analysis. Trying to fit these neurons in the Kupfermann and Weiss box of a command neuron may be the problem. It is not necessary to call these neurons command neurons for this work to be important. The experiments show that the PDN DNs are critically involved in the halt phase of chemotaxis, and PDN DNs provide a nexus for further circuit analysis and dissection of the behavioral mechanisms. Please re-frame your claims so that they are more nuanced and reflect the actual complexity of the whole system.

3) A synthesizing diagram in the manuscript that can tie in all of the observations (anatomical, functional, and behavioral) is missing. Even just a cartoon schematic would go a long way in making this work more easily digestible by all readers, including non-experts. This schematic could also effectively raise clearly the outstanding information for further experiments to determine the circuit mechanisms whereby PDM-DN and its downstream partners control halting and head-casting behaviors. What are the take-home messages to neuroscientists interested in behavioral control? We encourage the authors to provide such a synthesis in the final, published manuscript.

---

## [Author Response]

Essential revisions:1) Fundamental for the interpretation of all the results presented in the paper is the demonstration that the PDM-DNs plays a role in chemotaxis and not in crawling itself. The authors should show that when there is no odor and PDM-DNs are blocked, the larvae behave as their parental genetic controls. The experiment in Figure 3—figure supplement 1 no odor, should be performed with the same number of animals (30) required to see a significant difference in the chemotaxis experiment. If the behavior is not as the controls, then the entire paper will have to be reinterpreted and rewritten.

We have conducted the additional experiments requested by the reviewers. In a new dataset, statistical comparisons of no-odor controls shown in Figure 3—figure supplement 1 are based on a sample size of 31. The difference between the median of the test and control conditions is not statistically significant (p-value for turn rate= 0.2617; p-value for run speed= 0.1857). These results strengthen our conclusion that silencing the activity of PDM-DN does not result in locomotor defects that would make larvae crawl slower or turn less frequently.

2) The sufficiency and necessity tests performed and the associated analyses are very impressive, and the data are convincing; however, they do not show necessity for halts or for head casts. Rather, they show that the PDN DNs play a critical role in halts and head casts associated with down-gradient runs but are not essential for such halts and head casts. Similarly, the sufficiency tests do not present an absolute result indicating that activation of the neuron produces a halt. It is not important to claim necessity or sufficiency but rather to present the impressive circuit analysis. Trying to fit these neurons in the Kupfermann and Weiss box of a command neuron may be the problem. It is not necessary to call these neurons command neurons for this work to be important. The experiments show that the PDN DNs are critically involved in the halt phase of chemotaxis, and PDN DNs provide a nexus for further circuit analysis and dissection of the behavioral mechanisms. Please re-frame your claims so that they are more nuanced and reflect the actual complexity of the whole system.

We are thankful to the editor and reviewers for encouraging us to reevaluate the appropriateness of the concept of command neuron as proposed by Kuperfmann and Weiss (1978). We agree that this framework is so restrictive that it might lead to unfortunate conclusions, as has been argued by Yoshihara and Yoshihara (2018). We reformulated our conclusions accordingly.

3) A synthesizing diagram in the manuscript that can tie in all of the observations (anatomical, functional, and behavioral) is missing. Even just a cartoon schematic would go a long way in making this work more easily digestible by all readers, including non-experts. This schematic could also effectively raise clearly the outstanding information for further experiments to determine the circuit mechanisms whereby PDM-DN and its downstream partners control halting and head-casting behaviors. What are the take-home messages to neuroscientists interested in behavioral control? We encourage the authors to provide such a synthesis in the final, published manuscript.

We thank the reviewers for this suggestion. In the new Figure 7, we added a schematic diagram that illustrates the sensorimotor pathway anchored to the PDM-DN neuron. We summarized the results of our study about the function of the pathway and we highlighted the points that were left to speculation.